# Adaptive Ketogenic–Mediterranean Protocol (AKMP) in Real Clinical Practice: 14-Week Pre–Post Cohort Study on Glucolipid Markers and Safety

**DOI:** 10.3390/nu17223559

**Published:** 2025-11-14

**Authors:** Cayetano García-Gorrita, Nadia San Onofre, Juan F. Merino-Torres, Jose M. Soriano

**Affiliations:** 1Food & Health Lab, Institute of Materials Science, University of Valencia, 46980 Paterna, Spain; cayeggf@gmail.com; 2NUTRALiSS Research Group, Faculty of Health Sciences, Universitat Oberta de Catalunya, Rambla del Poblenou 156, 08018 Barcelona, Spain; nsan_onofre@uoc.edu; 3Joint Research Unit on Endocrinology, Nutrition and Clinical Dietetics, University of Valencia-Health Research Institute La Fe, 46026 Valencia, Spain; merino_jfr@gva.es; 4Department of Medicine, Faculty of Medicine, University of Valencia, 46010 Valencia, Spain; 5Department of Endocrinology and Nutrition, University and Polytechnic Hospital La Fe, 46026 Valencia, Spain

**Keywords:** Adaptive Ketogenic–Mediterranean Protocol (AKMP), Ketogenic Mediterranean Diet, nutritional ketosis, insulin resistance (HOMA-IR), remnant cholesterol, triglyceride-glucose index (TyG), triglyceride-to-HDL cholesterol ratio, metabolic adaptation (anti-plateau algorithm), body composition, obesity, cardiometabolic risk, low-carbohydrate diet

## Abstract

Background/Objectives: Overweight and obesity are associated with insulin resistance, atherogenic dyslipidemia, and low-grade inflammation. We evaluated analytical safety and within-person metabolic changes under the Adaptive Ketogenic–Mediterranean Protocol (AKMP) in real-world practice. Methods: Single arm, prospective pre–post cohort. We enrolled 112 adults; 105 completed 14 weeks of AKMP (12 in nutritional ketosis ≤ 20 g carbohydrate/day + 2 of gradual reintroduction). Fasting venous samples were analyzed in accredited laboratories (glycolipid profile, hepatic–renal function, inflammatory markers; insulin, thyroid hormones, cortisol). HOMA-IR, TyG, and remnant cholesterol (RC) were calculated; body composition was measured by segmental bioimpedance. Paired analyses were used, with hierarchical gatekeeping for the conditional co-primary outcome and prespecified Δ~Δ correlations. Results: HOMA-IR −52.8% (Δ −1.80; *p* < 0.001) and RC −35.1% (Δ −10.64 mg/dL; *p* < 0.001); fasting glucose −13.7 mg/dL, insulin −5.9 μU/L; TyG −0.23 and TG/HDL-c −1.21 (all *p* < 0.001). Lipids: TG −35.1% and LDL-c −11.2%; HDL-c remained stable. Anthropometry: weight −14.85 kg (−14.7%) and trunk fat −4.88 kg (−22.2%) (*p* < 0.001). Safety: no serious adverse events; GGT −47.0%, eGFR +11.0%, and CRP −24.6% (*p* < 0.001). Prespecified correlations supported the internal consistency of the glycolipid axis (e.g., ΔHOMA-IR~ΔTyG; ΔRC~ΔHOMA-IR). Conclusions: In adults with overweight or obesity, the AKMP was associated with improvements in the glucose–insulin axis, atherogenic profile (RC, TG/HDL-c, TG), and body composition, while maintaining a favorable safety profile. The protocol appears feasible in clinical practice and monitorable with routine laboratory tests, although randomized controlled trials are needed to confirm causality and long-term sustainability.

## 1. Introduction

The global rise in obesity is closely associated with a cardiometabolic profile characterized by insulin resistance (IR), central adiposity with trunk predominance, atherogenic dyslipidemia, and low-grade systemic inflammation. Collectively, this pathophysiological cluster accounts for a substantial portion of contemporary cardiometabolic risk. Experimental and clinical evidence has established metabolic inflammation in adipose tissue—including cytokine signaling such as TNF-α and immune cell infiltration—as a key determinant of IR, β-cell dysfunction, and progression to type 2 diabetes (T2D) [1,2].

In clinical practice, the atherogenic dyslipidemia typically seen in obesity/overweight with IR and metabolic syndrome is characterized by elevated triglycerides (TG), low HDL c, and small, dense LDL particles. Among readily available composite indices, two are particularly relevant: (i) TG/HDL C, a simple marker of IR and cardiometabolic risk, and (ii) the triglyceride–glucose index (TyG), defined as TyG = ln[TG (mg/dL) × glucose (mg/dL)/2] [3,4]. In addition, remnant cholesterol (RC), calculated as TC − LDL C − HDL C, has emerged as a causal factor in atherosclerosis and low-grade inflammation, supported by observational, genetic, and mechanistic data [5].

The glucose–insulin axis remains central both for its pathophysiological relevance and for its straightforward assessment in clinical settings. In both research and practice, IR is commonly estimated using fasting-derived indices: HOMA IR = glucose (mg/dL) × insulin (µU/mL)/405, and TyG. These measures show good concordance with gold-standard techniques such as the hyperinsulinemic euglycemic clamp and are particularly suitable for pre–post comparisons due to their low cost and practicality [4,6,7,8]. From a dietary-intervention standpoint, carbohydrate-restricted diets—including well-formulated ketogenic variants—consistently lower triglycerides and can achieve weight loss comparable to low-fat comparators [9]. However, long-term between-diet differences are small and context-dependent, and most randomized trials vary in energy prescription and seldom verify sustained ketosis biochemically; therefore, diet-specific effects under strictly isocaloric, ketosis-verified conditions cannot be inferred [9]. In energy-matched settings, differences in weight loss are typically modest [10]. In turn, the classic Mediterranean diet—rich in extra-virgin olive oil (EVOO), nuts, fish, and vegetables—reduces major cardiovascular events in primary prevention in high-risk populations, as shown by Estruch and colleagues in the PREDIMED study [11]; moreover, in VLED-induced mild ketosis (fasting β-OHB ≈ 0.5 mmol/L), Sumithran et al. reported suppression of the usual ghrelin rise and lower appetite ratings during ketosis compared with the same participants after refeeding [12]. This background supports the rationale of the Adaptive Ketogenic–Mediterranean Protocol (AKMP) [13]: to combine the glycolipid efficacy and appetite control (including the possible mitigation of anxiety and addictive patterns associated with continued consumption of refined carbohydrates and ultra-processed foods) linked to ketosis, with Mediterranean cardioprotective attributes (EVOO, oily fish, non-starchy vegetables, nuts), seeking clinical feasibility and a favorable safety profile.

Here, “ketogenic–Mediterranean” denotes a time-sequenced approach: a ketogenic induction implemented with Mediterranean food quality (e.g., extra-virgin olive oil as the primary culinary fat, abundant non-starchy vegetables, preference for fish and lean proteins, and nuts), followed by progressive carbohydrate reintroduction and an individualized Mediterranean maintenance phase. The maintenance phase is not low-carbohydrate by design; carbohydrate intake exceeded ~100 g/day as a minimum reference and was then individualized upward according to energy expenditure, physical activity, and clinical context, with adherence assessed by food-based indices rather than fixed macronutrient quotas.

Consistent with this, Gardner and colleagues, in the crossover Keto-Med study in individuals with prediabetes/T2D, observed similar improvements in HbA1c with Mediterranean and low-carbohydrate ketogenic-type diets; TG reduction was greater with the ketogenic diet, with differences in LDL-C, findings that support combined and personalized strategies [14]. A key component of the AKMP is its anti-plateau algorithm, designed to prolong periods of active weight loss, mitigate metabolic adaptation, and promote clinical adherence, for which sustaining patient motivation is crucial [15,16,17]. Under plateaus of ≥14 days with documented ketosis, the protocol acts in a personalized manner: (a) if hunger predominates and ketosis has not achieved a sufficient anorexigenic effect, despite β-hydroxybutyrate (β-OHB) levels ≥ 0.6 mmol/L and a persistent sensation of hunger reported by the patient, protein is increased to 1.2–1.6 g/kg of ideal body weight; (b) if satiety is adequate, a modest reduction of 100–200 kcal/d is proposed at the expense mainly of fat, preserving protein (per kg of body weight per day). This strategy rests on two pillars: (i) the thermodynamic advantage described with very low-carbohydrate patterns—with increases in total energy expenditure (TEE) during weight-loss maintenance—and (ii) the anorexigenic effect of ketosis and protein prioritization when the ketotic signal is not sufficient, which adds satiety and a greater thermic effect without compromising ketosis [12,18,19,20,21,22].

Moreover, controlled-feeding studies suggest the existence of a “metabolic advantage” in very low-carbohydrate patterns whereby, after ≈2–3 weeks of adaptation, a modest but consistent increase in TEE is observed, with a dose–response relationship of ≈+50 kcal/d per each −10% of carbohydrates and robust findings in prolonged trials assessed by doubly labeled water (DLW) [18,19,23]. From a physiological perspective, this pattern can be explained by additional energetic costs (gluconeogenesis, thermic effect of protein, and futile cycles—lipolysis/re-esterification TAG/FFA; Ca^2+^ cycling by the SERCA Ca^2+^-ATPase; creatine/phosphocreatine cycle) and by slightly lower efficiency of oxidative phosphorylation when fatty-acid oxidation predominates (greater electron flux via complex II and ETF:QO versus complex I, along with redox slip and proton leak), which reduces the ATP/O ratio and compels an increase in O_2_ consumption to maintain Δ ψ and ATP synthesis; in adipose tissue, the TAG/FFA cycle also consumes ATP and contributes to energy dissipation [24,25,26,27,28,29,30,31].

In the inflammatory and hormonal domain, Saslow and colleagues reported glycemic improvements and medication reduction in people with T2D/prediabetes subjected to a very low-carbohydrate diet versus a moderate-carbohydrate diet [32]. In line with inflammatory markers, the EMIKETO study (RCT in patients with migraine and overweight/obesity) observed decreases in CRP with a very low-calorie ketogenic diet (VLCKD) in a controlled setting [33], and a recent meta-analysis of randomized studies reported a significant reduction in CRP (with a non-significant downward trend in IL-6) in adults with overweight/obesity on ketogenic diets [34]. In the aforementioned 20-week study, ghrelin and leptin concentrations were lower with the lowest-carbohydrate diet, consistent with a hormonal milieu potentially more favorable for weight control [18].

In perspective, the evidence suggests consistent changes and a potential for metabolic improvement under well-formulated ketogenic approaches, recognizing interindividual variability and that areas of debate persist regarding effect magnitude and short-term response heterogeneity.

Finally, in phase 3 of the AKMP, metabolic flexibility and the Randle cycle are particularly relevant. As carbohydrate intake increases, leading to a corresponding rise in insulin signaling, malonyl-CoA levels rise, CPT-I is inhibited, and mitochondrial fatty acid entry decreases, shifting oxidation toward glucose. This elevates postprandial RQ and accounts for transient adjustments in the glucose–insulin axis during readaptation [31,35]. Jansen and colleagues documented via continuous glucose monitoring (CGM) and the oral glucose tolerance test (OGTT) that this glycemic readaptation after transitioning from low- to high-carbohydrate diets can last several weeks [36]. Accordingly, ADA guidelines recommend ≥150 g of carbohydrates/day during the 3 days prior to an OGTT to avoid false positives due to under-adaptation [37]. Therefore, the AKMP proposes a gradual reintroduction (≈50–100 g/day) prioritizing low glycemic load and high fiber (vegetables, berries, legumes in controlled portions) within a Mediterranean pattern with EVOO as the matrix fat, thereafter transitioning to a fully Mediterranean maintenance with carbohydrate intake > 100 g/day, individualized to energy needs and activity.

Taken together, these patterns primarily reflect a shift of oxidation toward fatty acids—within the framework of metabolic flexibility and glucose–fatty acid competition described by the Randle cycle—along with thermogenic adjustments [31,35].

Despite growing interest in ketogenic diets, few ≈14-week pre–post studies integrate a broad clinical panel of routine practice—hepatorenal and renal function (GGT, ALP, bilirubin, urea, creatinine, eGFR), magnesium, glucose/insulin, lipids with remnant cholesterol, TG/HDL-c and TyG, and CRP—together with body composition with emphasis on trunk fat (kg). This study addresses that gap by evaluating, in real-world clinical practice, the AKMP as a hypothesis-generating intervention, with prespecified Δ~Δ analyses among key variables (ΔHOMA-IR~Δtrunk fat; ΔHOMA-IR~ΔTyG; ΔGGT~Δ [TG/HDL-c]; ΔCRP~Δtrunk fat; Δremnant cholesterol~ΔHOMA-IR). Accordingly, our objective was to describe, over ≈14 weeks, analytical safety and the evolution of cardiometabolic parameters—including glycolipid markers—and body composition under the AKMP in adults with overweight or obesity. Primary outcome: ΔHOMA-IR (Post − Pre); conditional co-primary (gatekeeping): Δremnant cholesterol (ΔRC); secondary outcomes: TyG, TG/HDL-c, lipid profile (CT, HDL-c, LDL-c, TG), hepatorenal function, magnesium, CRP, thyroid axis (TSH, FT4, FT3, FT3/FT4), basal cortisol, and anthropometry/composition (BMI, weight, fat mass, and trunk fat).

## 2. Materials and Methods

### 2.1. Study Design and Ethical Considerations

We conducted a prospective cohort study with a 14-week pre–post design (12 weeks in nutritional ketosis followed by 2 weeks of gradual carbohydrate reintroduction until achieving an individualized balance approximating a Mediterranean pattern, typically >100 g CH/d). Recruitment took place between June and August 2024 at a private clinic with an in-house Clinical Analysis Laboratory and Dietetics and Nutrition Unit in Real de Gandía (Valencia, Spain). A total of 112 adults expressed interest; 105 completed the intervention (7 withdrew due to lack of adherence to the protocol); see Figure 1 (participant flowchart, STROBE). Analyses were conducted on complete pairs (pre–post), and no missing values were imputed.

The sample was one of convenience (*n* = 105); no formal sample size calculation was performed.

The protocol was approved by the Human Research Ethics Committee of the University of Valencia (ref. 2023 MED 2718369; certificate 6 June 2023). The registered application (REGAGE23e00029679323, 9 May 2023) detailed pseudonymization/coding of personal data, the use of human biological samples, and the collection of written informed consent. Confidentiality was ensured in accordance with the General Data Protection Regulation (GDPR), and the study was conducted in line with the Declaration of Helsinki.

### 2.2. Participants

**Setting.** Recruitment took place at a private clinic with its own clinical laboratory; biochemical determinations were supervised by a clinical analysis specialist. Hormonal assays (TSH, FT3, FT4, insulin, and cortisol) were performed at Laboratorio Juan Bta. Montoro, S.L., accredited under ISO 15189 [38] and certified ISO 9001 [39], following their quality procedures and manufacturer instructions (see Section 2.5 and Appendix A). The dietary intervention and follow-up were directed by a licensed dietitian-nutritionist (principal investigator).

**Recruitment method.** Consecutive patients were enrolled when seeking weight loss or metabolic health improvement, some referred by primary care physicians or regional hospitals in Valencia and Alicante. No public or external recruitment was undertaken.

At the first visit, all participants received a detailed explanation of the study and signed written informed consent describing the study objectives, dietary intervention (AKMP), collection and handling of biological samples, potential benefits and risks, and the voluntary nature of participation, including the right to withdraw at any time without consequences. Confidentiality was guaranteed through data coding in accordance with GDPR; each participant was assigned a unique code for registration and analysis.

**Inclusion criteria.** Adults (≥18 years) presenting with weight loss and/or cardiometabolic health improvement as goals; ability to comply with the AKMP for 14 weeks (biweekly visits and telematic submissions); stable chronic medication ≥ 3 months; ability to report weight and ketonuria remotely; provision of informed consent and acceptance of data coding.

**Exclusion criteria.** Inborn errors of fatty acid metabolism (e.g., CPT I/II deficiency, carnitine–acylcarnitine translocase deficiency, or other β-oxidation defects), pyruvate carboxylase deficiency, or acute intermittent porphyria; type 1 diabetes; treatment with SGLT-2 inhibitors during ketogenic phases; acute pancreatitis; chronic kidney disease; severe hepatic failure; pregnancy or lactation; pacemakers or other electronic implants (contraindicated for BIA); age < 18 years.

### 2.3. Dietary Intervention

**Dietary protocol.** The nutritional protocol was the Adaptive Ketogenic–Mediterranean Protocol (AKMP) [13], designed to induce and maintain nutritional ketosis within a Mediterranean-style food matrix, with dynamic adjustments to prevent metabolic adaptation and address weight-loss plateaus. In AKMP, nutritional ketosis was induced primarily via carbohydrate restriction within a Mediterranean food matrix; very-low-calorie prescriptions were not used as the primary means to induce ketosis, and total energy was individualized with small, tailored adjustments only under documented weight-loss plateaus. The intervention lasted 14 weeks: 12 weeks in nutritional ketosis and 2 weeks of gradual carbohydrate reintroduction. The dietary prescription emphasized extra-virgin olive oil (EVOO), oily fish, nuts, and non-starchy vegetables, while restricting foods rich in starch and added sugars. Initial macronutrient goals were ≤20 g/d digestible (net) carbohydrates, 20–40% protein, and 50–70% fat of total energy intake. Nutritional ketosis was operationally defined as capillary β-hydroxybutyrate (β-OHB) ≥ 0.5 mmol/L, a target monitored and reinforced throughout the first 12 weeks. During the induction phase (12 weeks), carbohydrate intake was fixed at ≤20 g/day of digestible (net) carbohydrates to achieve and maintain nutritional ketosis; capillary β-hydroxybutyrate (β-OHB) was measured every 15 days during clinic visits and complemented by photographic documentation of urinary ketones between visits. Protein was initially prescribed at 20–40% of total energy intake and, under ≥14-day weight-loss plateaus with documented ketosis, was adjusted on an individualized basis (e.g., increased up to ~1.2–1.6 g/kg/day of ideal body weight if persistent hunger was reported despite β-OHB ≥ 0.6 mmol/L; or total energy was reduced by ~100–200 kcal/day primarily at the expense of fat when satiety was adequate), while maintaining ≤20 g/day carbohydrates throughout the ketogenic phase. Personalized menus and portions were prepared using Dietopro^®^ (Dietopro Software, Puerto de Sagunto, Valencia, Spain; web-based platform, dietopro.com, accessed September–December 2024), respecting individual preferences and tolerances.

Resting energy expenditure was estimated using the Mifflin–St Jeor equation [40], and total energy expenditure was obtained by applying an activity factor. In cases of weight plateau, adjustments were guided by patient-reported information (e.g., hunger/satiety, cravings, adherence, schedules, activity level). Under ≥14-day weight-loss plateaus with documented ketosis, adjustments were individualized: protein was increased up to ~1.2–1.6 g/kg/day of ideal body weight when persistent hunger was reported despite β-OHB ≥ 0.6 mmol/L, or total energy was reduced by ~100–200 kcal/day primarily from fat when satiety was adequate; carbohydrate intake remained ≤20 g/day throughout the ketogenic phase. Personalization was based on an agreed target weight and realistic intermediate goals; caloric–protein adjustments were individualized according to weight trajectory, satiety, and feasibility, applying the individualized anti-plateau strategies described above. The intervention was delivered by a licensed dietitian-nutritionist with 7 years of clinical experience in ketogenic and Mediterranean diets. Support was provided through individualized in-person consultations, including tailored nutrition education and practical problem-solving. Consultations were conducted at a private health center equipped with Dietetics and Nutrition and Clinical Analysis units. Follow-up consisted of biweekly visits (8 sessions, 30–60 min each) over 14 weeks. During the last 2 weeks, carbohydrates were gradually reintroduced, as outlined by the AKMP, to achieve an individualized balance approximating a Mediterranean pattern (typically >100 g CH/d). The week 14 visit was scheduled with a ±7-day window. The intervention began during the week of 1–6 September 2024 (depending on participant assignment) and concluded on 13 December 2024. Any adjustments to the plan (energy, macronutrient distribution, number of meals, or specific foods) were documented in Dietopro^®^ to ensure traceability and consistency across sessions.

**Procedures and schedule.** At the initial visit, the dietitian-nutritionist provided an individual explanation of the AKMP procedure (phases, macronutrient targets, and visit schedule) and delivered a rules document along with the first closed, personalized menu (weighed portions, no food changes except permitted substitutions). Visits were scheduled every 2 weeks over the 14-week intervention (≈8 sessions). When issues could not be resolved remotely, priority in-person visits were arranged.

Each participant attended consultations on the same day of the week and at approximately the same time throughout follow-up, in order to standardize measurements (weight, BIA, capillary ketones) and reduce circadian variability. If participants were unable to attend at their usual time, Saturday sessions were available as a recovery option.

**Operational rules and permitted substitutions.** Menus were individualized and closed; participants were instructed to consume all assigned portions and avoid changes without prior authorization. Allowed modifications were limited to: (i) rearranging meals within the same day (while maintaining all portions); (ii) swapping complete days within the weekly plan (condition: all 7 days completed by week’s end); and (iii) type-for-type substitutions of equal weight: red meat ↔ red meat, white meat ↔ white meat, oily fish ↔ oily fish, and lean fish ↔ lean fish (with some tolerance for bone weight in fish). One glass of wine per day was permitted on request to support social adherence; all other alcoholic beverages were prohibited. Any additional changes required prior consent from the dietitian and documentation in Dietopro^®^.

Participants were instructed to maintain stable medication and supplements throughout the 14 weeks; any changes had to be reported immediately and recorded in clinical documentation.

**Monitoring and adherence.** Patients reported fasting weight twice per week and submitted photographs of urine ketone test strips via (WhatsApp Business (Meta Platforms, Inc., Menlo Park, CA, USA; approx. version 2.24.19.82–2.24.23.78, accessed September–December 2024).; Ketostix^®^ strips (Bayer, Leverkusen, Germany) were provided at each visit. Capillary β-OHB was measured every 15 days during visits using a portable meter with specific test strips. Progress was reviewed at each visit, and any adjustments to energy intake, macronutrient distribution, or number of meals were documented in Dietopro^®^. Safety monitoring, including biweekly anamnesis, a 24/7 contact channel, and biomarker assessments at baseline and week 14, is detailed in Section 2.11.

**Protocol fidelity.** Fidelity was assessed by cross-checking electronic messages/files and weight and ketonuria records against prescribed targets, with plan updates as needed. No patient-completed checklists were used; verification was performed through clinical and documentary review (WhatsApp + Dietopro^®^).

A comprehensive operational description of the AKMP, together with mapping of the 12 TIDieR items (materials, procedures, who, how/where, when/how much, tailoring, modifications, and fidelity—planned and actual), is provided in Appendix A.

### 2.4. Collection and Handling of Biological Samples

Fasting blood samples were collected in serum separator tubes, allowed to clot for 20–30 min, and centrifuged within ≤2 h. Each sample was assigned a pseudonymized code. Collections were performed between 08:00 and 11:00 at the accredited laboratory. After centrifugation and aliquoting, samples were stored at −80 °C in an ultra-freezer (−86 °C, UC ULT U100, PharmaFred, Lleida, Spain) and analyzed within 12 months. For hormonal assays (TSH, FT3, FT4, insulin, cortisol), coded serum aliquots were shipped on dry ice (≤−78 °C) to Laboratorio Juan Bta. Montoro, S.L. (Valencia, Spain, ISO 15189/ISO 9001). Chain of custody, reagent/calibrator batches, and processing dates were documented. Analyses were performed on the ADVIA Centaur^®^ XP platform (Siemens Healthineers, Erlangen, Germany), following the manufacturer’s IFU and internal QC procedures (≥2 levels/day and after lot changes/maintenance).

### 2.5. Biochemical Determinations

**Biochemical panel (serum)**: glucose, total cholesterol, HDL-c, triglycerides, C-reactive protein (standard CRP, not hs-CRP), γ-glutamyltransferase (GGT), alkaline phosphatase, total bilirubin, urea, creatinine, uric acid, and magnesium. Analyses were performed on a BS200E automatic analyzer (Mindray Bio-Medical Electronics Co., Ltd., Shenzhen, China), based on absorbance photometry and turbidimetry, using validated commercial reagents (Spinreact, Sant Esteve de Bas, Girona, Spain; BioSystems, Barcelona, Spain).

**Sample centrifugation:** serum/plasma separation was performed with a Unicen 21 universal ventilated centrifuge (Ortoalresa, Madrid, Spain), equipped with microprocessor-controlled RCF (× *g*)/rpm and time programming, and compatible with swinging and fixed-angle rotors for 10–15 mL tubes. Operating conditions: 3000–4000 rpm for 10–15 min at room temperature (≈20–22 °C); equivalent RCF values depending on rotor, within equipment specifications (max. 2938× *g* at 4200 rpm). Runs were conducted according to the laboratory SOP (rotor selection, braking), with load balancing, moderate braking to avoid resuspension, and aseptic handling. Serum was separated immediately, aliquoted, and stored at −80 °C or analyzed according to protocol.

**Hormonal panel (TSH, FT3, FT4, insulin, cortisol):** performed at Laboratorio Juan Bta. Montoro, S.L. (ISO 15189/ISO 9001) using direct chemiluminescence immunoassays on the ADVIA Centaur^®^ XP system (Siemens Healthineers, Erlangen, Germany; TSH3-Ultra II, FT3, FT4, Insulin, Cortisol). Manufacturer IFU was strictly followed, with calibration according to recommended schedules and QC at ≥2 levels per analytical day and after lot or maintenance changes. Control ranges, imprecision claims, and intra-/inter-assay %CV are provided in Appendix A (laboratory QC).

**Capillary ketonemia (β-OHB):** measured at clinic visits every 15 days with GlucoMen^®^ areo GK (A. Menarini Diagnostics, Florence, Italy) using single-use GlucoMen^®^ areo β Ketone Sensor strips. Electrochemical method with β-hydroxybutyrate dehydrogenase (mediator 1,10-phenanthroline-5,6-dione); sample volume 0.8 µL; analysis time 8 s; range 0.1–8.0 mmol/L; hematocrit 20–60%; “HI” flag above range. The system was calibrated against Stanbio β-Hydroxybutyrate LiquiColor^®^ No. 2440 (EKF Diagnostics–Stanbio Laboratory, Boerne, TX, USA). Determinations followed the manufacturer’s IFU and QC plan, using GlucoMen^®^ areo Ket Control when appropriate.

**Quality control (QC):** intra- and inter-series coefficients of variation were documented for each analyte (two-level internal controls and acceptance criteria), along with control materials and calibrations. Full details are available in Appendix A (laboratory QC).

**Internal control and imprecision verification:** internal controls were run at two levels (normal and pathological), and %CV values for repeatability (within-run/within-assay/within-day) and intermediate precision (between-run/between-assay/between-day) were estimated. For the biochemical panel (BS200E; Spinreact/BioSystems), QC was performed at the study laboratory; for the hormonal panel (TSH, FT3, FT4, insulin, cortisol; ADVIA Centaur^®^ XP), QC was performed at Laboratorio Juan Bta. Montoro, S.L. (ISO 15189/ISO 9001) according to IFU. Control materials/calibrators, manufacturer imprecision claims, and internal %CV (mean ± SD and %CV by condition) are detailed in Appendix A (laboratory QC), together with the applied calibration/verification plan.

### 2.6. Derived Indices

**HOMA-IR**: [insulin (µU/mL) × glucose (mg/dL)]/405.

**TyG:** ln[TG (mg/dL) × fasting glucose (mg/dL)/2] [41].

**Remnant cholesterol:** Total cholesterol − LDL-c − HDL-c.

**Calculated LDL-c:** Friedewald formula (LDL-c = TC − HDL-c − TG/5) when TG < 400 mg/dL [42]; for the single participant with TG ≥ 400 mg/dL, the Sampson equation was applied [43].

**eGFR:** CKD-EPI (race-free) equation:eGFR = 142 × min(Scr/k, 1)^a^ × max(Scr/k, 1)^−1.200^ × 0.9938^age^ × [1.012 if female]
where k = 0.7 (female) or 0.9 (male), and a = −0.241 (female) or −0.302 (male) [44].

### 2.7. Units and Derived Variables (Thyroid Panel)

For comparability with the literature, FT3 and FT4 were also reported in SI units from original determinations: FT3 (pmol/L) = FT3 (pg/mL) × 1.536; FT4 (pmol/L) = FT4 (ng/dL) × 12.87. The FT3/FT4 ratio was treated as a dimensionless variable and calculated per participant using SI values.

### 2.8. Body Composition (BIA)

Body composition was assessed at baseline and week 14 using direct segmental multifrequency bioelectrical impedance analysis (DSM-BIA) with InBody 270 (InBody Co., Ltd., Seoul, Republic of Korea). The device uses 8-point tetrapolar electrodes and measures impedance at two frequencies (20 and 100 kHz) across five segments (right/left arm, trunk, right/left leg); test time ≈ 15 s. Manufacturer’s conditions were followed: skin–electrode contact, removal of shoes/accessories, clean palms/soles, standing ~5 min, fasting ≥ 2 h, no prior exercise, voiding if needed, and avoiding evaluator contact during measurement. No BIA was performed in participants with pacemakers or implants; the test is not recommended in pregnancy.

### 2.9. Statistical Analysis

The analysis was prespecified before data review. The primary outcome was ΔHOMA-IR (two-tailed, α = 0.05). A conditional co-primary (gatekeeping), Δremnant cholesterol, was evaluated only if the primary was significant. Secondary outcomes are reported with nominal *p*-values (no multiplicity adjustment).

For each variable, baseline and post-intervention means ± SD were calculated, along with absolute change (Δ = Post − Pre) and relative change (%Δ = 100 × (Post − Pre)/Pre). Pre–post analyses were conducted on complete pairs (complete-case; no imputation).

Paired *t*-tests or Wilcoxon signed-rank tests were applied depending on data distribution. Analyses were stratified by sex; differences in Δ between sexes were tested with independent two-tailed *t*-tests. For variables with more than two categories, ANOVA was used.

Effect sizes were expressed as Cohen’s d with 95% CI. In pre–post and sex-stratified analyses, d is presented either as magnitude (absolute value) or with sign depending on software output. In all cases, the direction of effect is interpreted via Δ and %Δ. Unless stated otherwise, tests were two-tailed (α = 0.05).

Prespecified bivariate correlations among changes (Δ–Δ) were assessed with Pearson coefficients (two-tailed, α = 0.05), reporting r and *p*. Exact *p*-values are reported to three decimals, or as *p* < 0.001 where appropriate; statistical significance was set at *p* < 0.05.

Rounding policy. Δ and %Δ were computed from non-rounded means, then rounded (Δ to two decimals; %Δ to one). Discrepancies up to ±1.0 percentage points may occur when recalculating from displayed values.

Analyses were performed with IBM SPSS Statistics v27 (IBM Corp., Armonk, NY, USA).

### 2.10. Quality Control

**Clinical laboratory.** For the biochemical panel, the study laboratory employed two-level internal controls and documented intra- and inter-assay %CVs; for the hormonal panel (TSH, FT3, FT4, insulin, cortisol), determinations and IQC were performed at Laboratorio Juan Bta. Montoro, S.L. (ISO 15189/ISO 9001). Full details (control materials, acceptance criteria, internal %CVs, and manufacturer claims) are provided in Appendix A.

**Data management.** Complete-case analyses were performed; no missing values were imputed. No replacements were planned; in the event of missing data, available cases would be analyzed and reasons documented. Extreme values were excluded only when there was pre-analytical evidence of error (e.g., hemolysis).

**Confidentiality.** Coding and pseudonymization were carried out in accordance with the ethics approval of the University of Valencia.

**Measurement bias reduction.** Laboratory personnel were blinded to the temporal condition of the sample (baseline vs. week 14) and to the study objectives.

**Reporting guidelines compliance.** The manuscript follows STROBE (cohorts), TREND (non-randomized interventions), and TIDieR (replicable intervention description). Complete checklists are provided as Appendix A. Analytical quality-control specifications are provided in Appendix A (laboratory QC).

### 2.11. Safety

Safety was assessed at each visit (every 2 weeks) through a focused clinical history and a 24/7 contact channel. Adverse events (AEs) and serious adverse events (SAEs) were prespecified; the risk window spanned from baseline to week 14. Sources included the clinical interview and medical records; denominators were the number of exposed participants and person-time at risk. Severity grading and causality were adjudicated using standardized clinical criteria. Concomitant medications and potential intercurrent events were tracked throughout follow-up; any relevant change in medication or concomitant intervention was documented. Safety biomarkers (hepatic, renal, inflammatory) were measured at baseline and week 14; results are presented in Section 3.4 (Safety and adverse events). Minor symptoms typical of the ketogenic phase (e.g., headache, cramps, “keto-flu”–like symptoms) were not systematically collected, as they were not a predefined safety endpoint in this study.

## 3. Results

### 3.1. Sample and Adherence

A total of 112 participants were assessed for eligibility; 105 completed the ≈14-week intervention (12 weeks of ketosis + 2 weeks of reintroduction) and were included in the final analysis, while 7 were excluded due to lack of adherence (see Figure 1). All analyses were performed on the full cohort (*n* = 105). The number of paired observations per variable is specified in each table. The mean age was 46.7 years (range 21–75), with 73% women (*n* = 77) and 27% men (*n* = 28). Baseline cohort characteristics are presented in Table 1. Adherence was verified as described in Methods (weight and urinary ketones reported twice weekly, and capillary ketonemia assessed biweekly during clinic visits).

### 3.2. Primary Outcome, Conditional Co-Primary, and Secondary Outcomes

**Primary outcome.** In the total cohort, HOMA-IR decreased by 52.8% (Δ = −1.80; *p* < 0.001; 95% CI for d: 0.708–1.168), reflecting a marked improvement in insulin resistance after 14 weeks of AKMP.

**Conditional co-primary.** Remnant cholesterol was reduced by 10.64 mg/dL (−35.1%; *p* < 0.001; 95% CI for d: 0.456–0.880) and was considered **confirmatory** under the prespecified gatekeeping hierarchy.

**Secondary outcomes.** Changes in the remaining predefined markers are summarized below and in Table 2, Table 3 and Table 4 (total cohort, women, and men, respectively), using two-tailed tests (α = 0.05). Fasting glucose decreased by 13.67 mg/dL (−13.9%; *p* < 0.001; 95% CI for d: 0.803–1.279), fasting insulin by 5.91 µU/L (−44.1%; *p* < 0.001; 95% CI for d: 0.758–1.227), the TyG index by 0.23 (−6.0%; *p* < 0.001; 95% CI for d: 1.014–1.529), and the TG/HDL-c ratio by 1.21 (−37.3%; *p* < 0.001; 95% CI for d: 0.349–0.760).

For anthropometric measures (*n* = 105 paired observations), body fat mass decreased by 12.06 kg (−26.4%; *p* < 0.001; 95% CI for d: 2.837–3.812), trunk fat by 4.88 kg (−22.2%; *p* < 0.001; 95% CI for d: 1.507–2.132), body weight by 14.85 kg (−14.7%; *p* < 0.001; 95% CI for d: 2.812–3.781), and BMI by 5.45 kg/m^2^ (−14.7%; *p* < 0.001; 95% CI for d: 2.919–3.918).

**Sex-stratified comparisons of Δ.** Changes were similar in women and men for HOMA-IR (*p* = 0.663), glucose (*p* = 0.165), insulin (*p* = 0.839), TyG (*p* = 0.107), TG/HDL-c (*p* = 0.196), remnant cholesterol (*p* = 0.171), body fat mass (*p* = 0.086), trunk fat (*p* = 0.877), and BMI (*p* = 0.396). The only exception was body weight, where the reduction was significantly greater in men (*p* = 0.010).

**Table 2 nutrients-17-03559-t002:** Total cohort (*n* = 105). Metabolic and anthropometric variables (paired).

Variable	*n* Pairs	Pre (Mean ± SD)	Post (Mean ± SD)	Δ (Post − Pre)	95% CI for d (Cohen)	%Δ	*p*	*p* Between Sexes (Δ)
HOMA-IR	105	3.40 ± 2.58	1.61 ± 1.06	−1.80	0.708–1.168	−52.8	<0.001	0.663
Glucose (mg/dL)	105	98.39 ± 16.23	84.72 ± 8.24	−13.67	0.803–1.279	−13.9	<0.001	0.165
Insulin (µU/L)	105	13.41 ± 8.54	7.50 ± 4.60	−5.91	0.758–1.227	−44.1	<0.001	0.839
TyG (ln[TG × Glu/2])	105	3.82 ± 0.24	3.59 ± 0.18	−0.23	1.014–1.529	−6.0	<0.001	0.107
TG/HDL-c	105	3.24 ± 2.48	2.03 ± 0.88	−1.21	0.349–0.760	−37.3	<0.001	0.196
Remnant cholesterol (mg/dL)	105	30.33 ± 18.17	19.68 ± 7.75	−10.64	0.456–0.880	−35.1	<0.001	0.171
Body fat mass (kg)	105	45.68 ± 13.53	33.62 ± 13.17	−12.06	2.837–3.812	−26.4	<0.001	0.086
Trunk fat (kg)	105	22.04 ± 4.89	17.15 ± 6.00	−4.88	1.507–2.132	−22.2	<0.001	0.877
Weight (kg)	105	100.84 ± 20.52	85.99 ± 18.12	−14.85	2.812–3.781	−14.7	<0.001	0.01
BMI (kg/m^2^)	105	37.00 ± 7.41	31.55 ± 6.59	−5.45	2.919–3.918	−14.7	<0.001	0.396

Notes. Mean ± SD; Δ = Post − Pre; %Δ = 100 × (Post − Pre)/Pre. 95% CI for d (Cohen) = confidence interval for the effect size. *p*: paired test (Student’s *t*-test or Wilcoxon) according to distribution. *p* between sexes (Δ): unpaired comparison of the change magnitude (Δ) between women (F) and men (M). Rounding note. Δ and %Δ were calculated from non-rounded means and then rounded (Δ to 2 decimals; %Δ to 1). Due to rounding, discrepancies of up to ±1.0 percentage point may appear when recalculating with displayed values. Abbreviations. HOMA-IR = homeostatic model assessment of insulin resistance; TyG = triglyceride–glucose index [ln(TG × Glu/2)]; TG/HDL-c = triglycerides-to-HDL cholesterol ratio; BMI = body mass index; SD = standard deviation.

Sex-stratified results for women are presented in Table 3, and those for men in Table 4.

In women (*n* = 77 paired observations), HOMA-IR decreased by 51.0% (Δ = −1.68; *p* < 0.001; 95% CI for d: 0.646–1.179). Fasting glucose decreased by 13.48 mg/dL (−13.8%; *p* < 0.001; 95% CI for d: 0.792–1.354), fasting insulin by 5.52 µU/L (−41.9%; *p* < 0.001; 95% CI for d: 0.677–1.216), TyG by 0.22 (−5.5%; *p* < 0.001; 95% CI for d: 0.960–1.560), TG/HDL-c by 1.06 (−34.4%; *p* < 0.001; 95% CI for d: 0.241–0.712), and remnant cholesterol by 9.79 mg/dL (−33.8%; *p* < 0.001; 95% CI for d: 0.356–0.841).

For anthropometry, body fat mass decreased by 11.52 kg (−24.6%; *p* < 0.001; 95% CI for d: 2.904–4.103), trunk fat by 4.39 kg (−20.1%; *p* < 0.001; 95% CI for d: 1.279–1.961), body weight by 14.21 kg (−14.6%; *p* < 0.001; 95% CI for d: 3.035–4.278), and BMI by 5.46 kg/m^2^ (−14.6%; *p* < 0.001; 95% CI for d: 2.871–4.059).

**Table 3 nutrients-17-03559-t003:** Women (*n* = 77). Metabolic and anthropometric variables (paired).

Variable	*n* Pairs	Pre (Mean ± SD)	Post (Mean ± SD)	Δ (Post − Pre)	95% CI for d (Cohen)	%Δ	*p*
HOMA-IR	77	3.30 ± 2.56	1.62 ± 1.11	−1.68	0.646–1.179	−51.0	<0.001
Glucose (mg/dL)	77	97.39 ± 15.59	83.91 ± 8.38	−13.48	0.792–1.354	−13.8	<0.001
Insulin (µU/L)	77	13.14 ± 8.69	7.63 ± 4.85	−5.52	0.677–1.216	−41.9	<0.001
TyG (ln[TG × Glu/2])	77	3.79 ± 0.23	3.58 ± 0.17	−0.22	0.960–1.560	−5.5	<0.001
TG/HDL-c	77	3.05 ± 2.53	2.00 ± 0.89	−1.06	0.241–0.712	−34.4	<0.001
Remnant cholesterol (mg/dL)	77	28.97 ± 18.65	19.18 ± 7.12	−9.79	0.356–0.841	−33.8	<0.001
Body fat mass (kg)	77	46.75 ± 14.50	35.23 ± 14.03	−11.52	2.904–4.103	−24.6	<0.001
Trunk fat (kg)	77	21.84 ± 4.80	17.45 ± 6.01	−4.39	1.279–1.961	−20.1	<0.001
Weight (kg)	77	97.64 ± 20.82	83.43 ± 18.89	−14.21	3.035–4.278	−14.6	<0.001
BMI (kg/m^2^)	77	37.36 ± 8.14	31.90 ± 7.28	−5.46	2.871–4.059	−14.6	<0.001

Notes. Mean ± SD; Δ = Post − Pre; %Δ = 100 × (Post − Pre)/Pre. 95% CI for d (Cohen) = confidence interval for the effect size. *p*: paired test (Student’s *t*-test or Wilcoxon) depending on distribution. Rounding note. Δ and %Δ were calculated from non-rounded means and then rounded (Δ to 2 decimals; %Δ to 1). Discrepancies of up to ±1.0 percentage point may occur when recalculating with displayed values. Abbreviations. HOMA-IR = homeostatic model assessment of insulin resistance; TyG = triglyceride–glucose index [ln(TG × Glu/2)]; TG/HDL-c = triglycerides-to-HDL cholesterol ratio; BMI = body mass index; SD = standard deviation.

In men (*n* = 28 paired observations), HOMA-IR decreased by 57.4% (Δ = −2.11; *p* < 0.001; 95% CI for d: 0.541–1.453). Fasting glucose decreased by 14.18 mg/dL (−14.0%; *p* < 0.001; 95% CI for d: 0.504–1.403), fasting insulin by 7.00 µU/L (−49.4%; *p* < 0.001; 95% CI for d: 0.635–1.583), TyG by 0.26 (−6.7%; *p* < 0.001; 95% CI for d: 0.794–1.809), TG/HDL-c by 1.63 (−43.2%; *p* < 0.001; 95% CI for d: 0.359–1.210), and remnant cholesterol by 13.00 mg/dL (−38.2%; *p* < 0.001; 95% CI for d: 0.436–1.312).

For anthropometry, body fat mass decreased by 13.58 kg (−31.8%; *p* < 0.001; 95% CI for d: 2.340–4.237), trunk fat by 6.22 kg (−27.6%; *p* < 0.001; 95% CI for d: 2.065–3.788), body weight by 16.60 kg (−15.1%; *p* < 0.001; 95% CI for d: 2.099–3.842), and BMI by 5.43 kg/m^2^ (−15.1%; *p* < 0.001; 95% CI for d: 2.308–4.185).

**Table 4 nutrients-17-03559-t004:** Men (*n* = 28). Metabolic and anthropometric variables (pairs).

Variable	*n* Pairs	Pre (Mean ± SD)	Post (Mean ± SD)	Δ (Post − Pre)	95% CI of d (Cohen)	%Δ	*p*
HOMA-IR	28	3.68 ± 2.67	1.57 ± 0.93	−2.11	0.541–1.453	−57.4	<0.001
Glucose (mg/dL)	28	101.14 ± 17.89	86.96 ± 7.53	−14.18	0.504–1.403	−14.0	<0.001
Insulin (µU/L)	28	14.16 ± 8.20	7.17 ± 3.90	−7.00	0.635–1.583	−49.4	<0.001
TyG (ln[TG × Glu/2])	28	3.88 ± 0.24	3.62 ± 0.19	−0.26	0.794–1.809	−6.7	<0.001
TG/HDL-c	28	3.77 ± 2.32	2.14 ± 0.82	−1.63	0.359–1.210	−43.2	<0.001
Remnant cholesterol (mg/dL)	28	34.06 ± 16.50	21.06 ± 9.28	−13.00	0.436–1.312	−38.2	<0.001
Body fat mass (kg)	28	42.76 ± 10.06	29.18 ± 9.31	−13.58	2.340–4.237	−31.8	<0.001
Trunk fat (kg)	28	22.57 ± 5.19	16.35 ± 6.00	−6.22	2.065–3.788	−27.6	<0.001
Weight (kg)	28	109.65 ± 17.08	93.05 ± 13.77	−16.60	2.099–3.842	−15.1	<0.001
BMI (kg/m^2^)	28	36.03 ± 4.88	30.60 ± 4.07	−5.43	2.308–4.185	−15.1	<0.001

Notes. Mean ± SD; Δ = Post − Pre; %Δ = 100 × (Post − Pre)/Pre. 95% CI of d (Cohen) = effect-size interval (magnitude). *p*: paired test (Student’s *t*-test or Wilcoxon) according to distribution. Sex-stratified analysis was descriptive (no direct between-sex comparison). Rounding note. Δ and %Δ were calculated from non-rounded means and then rounded (Δ to 2 decimals; %Δ to 1). Due to rounding, discrepancies of up to ±1.0 percentage point may appear when recalculating with displayed values. Abbreviations. HOMA-IR = HOMA index of insulin resistance; TyG = triglyceride–glucose index [ln(TG × Glu/2)]; TG/HDL-c = triglycerides-to-HDL cholesterol ratio; BMI = body mass index; SD = standard deviation.

#### 3.2.1. Conventional Lipids

In the total cohort (*n* = 105), triglycerides decreased by 35.1% (Δ = −53.22 mg/dL; *p* < 0.001; 95% CI for d: 0.456–0.880). Total cholesterol decreased by 12.2% (Δ = −25.16 mg/dL; *p* < 0.001; 95% CI for d: 0.653–1.105). LDL-c decreased by 11.2% (Δ = −13.93 mg/dL; *p* < 0.001; 95% CI for d: 0.285–0.690), whereas HDL-c showed no significant change (Δ = −0.59 mg/dL; −1.2%; *p* = 0.508; 95% CI for d: −0.127–0.256).

In sex-stratified comparisons of the change (Δ), no significant differences were observed for triglycerides (*p* = 0.171), total cholesterol (*p* = 0.928), LDL-c (*p* = 0.865), or HDL-c (*p* = 0.484).

Conventional lipid changes for the total cohort are summarized in Table 5, with sex-stratified results for women and men in Table 6 and Table 7, respectively.

In women (*n* = 77), triglycerides decreased by 33.8% (Δ = −48.94 mg/dL; *p* < 0.001; 95% CI for d: 0.356–0.841). Total cholesterol decreased by 12.5% (Δ = −25.81 mg/dL; *p* < 0.001; 95% CI for d: 0.636–1.167). LDL-c decreased by 11.7% (Δ = −14.63 mg/dL; *p* < 0.001; 95% CI for d: 0.258–0.732). HDL-c showed no significant change (Δ = −1.39 mg/dL; −2.7%; *p* = 0.215; 95% CI for d: −0.082–0.367).

**Table 6 nutrients-17-03559-t006:** Conventional lipids—Women (*n* = 77).

Variable	*n* Pairs	Pre (Mean ± SD)	Post (Mean ± SD)	Δ (Post − Pre)	95% CI for d (Cohen)	%Δ	*p*
TG (mg/dL)	77	144.88 ± 93.06	95.94 ± 35.95	−48.94	0.356–0.841	−33.8	<0.001
TC (mg/dL)	77	206.05 ± 44.36	180.24 ± 35.41	−25.81	0.636–1.167	−12.5	<0.001
LDL-c (mg/dL)	77	125.10 ± 36.84	110.47 ± 28.73	−14.63	0.258–0.732	−11.7	<0.001
HDL-c (mg/dL)	77	52.00 ± 13.17	50.61 ± 10.73	−1.39	−0.082–0.367	−2.7	0.215

Notes. Mean ± SD; Δ = Post − Pre; %Δ = 100 × (Post − Pre)/Pre. 95% CI for d (Cohen). *p*: paired test (Student’s *t*-test or Wilcoxon) depending on distribution. LDL-c: Friedewald/Sampson according to TG (see Section 2.6). Sex-stratified analysis was descriptive (no direct comparison between sexes). Rounding note. Δ and %Δ were calculated from non-rounded means and then rounded (Δ to 2 decimals; %Δ to 1). Due to rounding, discrepancies of up to ±1.0 percentage point may occur when recalculating with displayed values. Abbreviations. TG = triglycerides; TC = total cholesterol; LDL-c = low-density lipoprotein cholesterol; HDL-c = high-density lipoprotein cholesterol; SD = standard deviation.

In men (*n* = 28), triglycerides decreased by 38.2% (Δ = −65.00 mg/dL; *p* < 0.001; 95% CI for d: 0.436–1.312). Total cholesterol decreased by 11.4% (Δ = −23.39 mg/dL; *p* < 0.001; 95% CI for d: 0.370–1.225). LDL-c decreased by 9.8% (Δ = −12.00 mg/dL; *p* = 0.021; 95% CI for d: 0.068–0.849). HDL-c showed a small, non-significant increase (Δ = +1.61 mg/dL; +3.3%; *p* = 0.219; 95% CI for d: −0.611–0.140).

**Table 7 nutrients-17-03559-t007:** Conventional lipids—Men (*n* = 28).

Variable	*n* Pairs	Pre (Mean ± SD)	Post (Mean ± SD)	Δ (Post − Pre)	95% CI for d (Cohen)	%Δ	*p*
TG (mg/dL)	28	170.29 ± 82.52	105.29 ± 46.41	−65.00	0.436–1.312	−38.2	<0.001
TC (mg/dL)	28	205.57 ± 31.69	182.18 ± 39.33	−23.39	0.370–1.225	−11.4	<0.001
LDL-c (mg/dL)	28	122.80 ± 29.22	110.80 ± 28.91	−12.00	0.068–0.849	−9.8	0.021
HDL-c (mg/dL)	28	48.71 ± 10.31	50.32 ± 11.26	1.61	−0.611–0.140	3.3	0.219

Notes. Mean ± SD; Δ = Post − Pre; %Δ = 100 × (Post − Pre)/Pre. 95% CI for d (Cohen). *p*: paired test (Student’s *t*-test or Wilcoxon) depending on distribution. LDL-c: Friedewald/Sampson according to TG (see Section 2.6). Sex-stratified analysis was descriptive (no direct comparison between sexes). Rounding note. Δ and %Δ were calculated from non-rounded means and then rounded (Δ to 2 decimals; %Δ to 1). Due to rounding, discrepancies of up to ±1.0 percentage point may occur when recalculating with displayed values. Abbreviations. TG = triglycerides; TC = total cholesterol; LDL-c = low-density lipoprotein cholesterol; HDL-c = high-density lipoprotein cholesterol; SD = standard deviation.

#### 3.2.2. Liver Function

In the total cohort, GGT decreased by 47.0% (Δ = −14.73 U/L; *p* < 0.001; 95% CI **for** d: 0.537–0.971). Alkaline phosphatase decreased by 6.9% (Δ = −10.73 U/L; *p* < 0.001; 95% CI for d: 0.202–0.599), and total bilirubin decreased by 18.2% (Δ = −0.11 mg/dL; *p* < 0.001; 95% CI for d: 0.165–0.560).

**Sex-stratified comparisons of Δ.** Differences were observed between women and men for GGT (*p* < 0.001) and total bilirubin (*p* = 0.002), whereas no difference was found for alkaline phosphatase (*p* = 0.356).

Liver function changes for the total cohort are presented in Table 8, with sex-stratified data for women and men in Table 9 and Table 10, respectively.

In women, GGT decreased by 47.6% (Δ = −11.12 U/L; *p* < 0.001; 95% CI for d: 0.438–0.935). Alkaline phosphatase decreased by 8.5% (Δ = −13.25 U/L; *p* < 0.001; 95% CI for d: 0.228–0.698). Total bilirubin decreased by 15.7% (Δ = −0.08 mg/dL; *p* = 0.006; 95% CI for d: 0.093–0.551).

**Table 9 nutrients-17-03559-t009:** Liver function—Women (*n* = 77).

Variable	*n* Pairs	Pre (Mean ± SD)	Post (Mean ± SD)	Δ (Post − Pre)	95% CI for d (Cohen)	%Δ	*p*
GGT (U/L)	77	23.34 ± 18.65	12.22 ± 6.95	−11.12	0.438–0.935	−47.6	<0.001
Alkaline phosphatase (U/L)	77	155.12 ± 38.90	141.87 ± 43.37	−13.25	0.228–0.698	−8.5	<0.001
Total bilirubin (mg/dL)	77	0.52 ± 0.27	0.44 ± 0.22	−0.08	0.093–0.551	−15.7	0.006

Notes. Mean ± SD; Δ = Post − Pre; %Δ = 100 × (Post − Pre)/Pre. 95% CI for d (Cohen). *p*: paired test (Student’s *t*-test or Wilcoxon) depending on distribution. Sex-stratified analysis was descriptive (no direct comparison between sexes). Rounding note. Δ and %Δ were calculated from non-rounded means and then rounded (Δ to 2 decimals; %Δ to 1). Due to rounding, discrepancies of up to ±1.0 percentage point may occur when recalculating with displayed values. Abbreviations. GGT = γ-glutamyltransferase; SD = standard deviation.

In men, GGT decreased by 46.2% (Δ = −24.68 U/L; *p* < 0.001; 95% CI for d: 0.551–1.468). Alkaline phosphatase showed no significant change (Δ = −3.82 U/L; −2.4%; *p* = 0.317; 95% CI for d: −0.183–0.565). Total bilirubin decreased by 23.0% (Δ = −0.17 mg/dL; *p* = 0.022; 95% CI for d: 0.066–0.846).

**Table 10 nutrients-17-03559-t010:** Liver function—Men (*n* = 28).

Variable	*n* Pairs	Pre (Mean ± SD)	Post (Mean ± SD)	Δ (Post − Pre)	95% CI for d (Cohen)	%Δ	*p*
GGT (U/L)	28	53.43 ± 29.82	28.75 ± 21.11	−24.68	0.551–1.468	−46.2	<0.001
Alkaline phosphatase (U/L)	28	158.50 ± 42.31	154.68 ± 43.89	−3.82	−0.183–0.565	−2.4	0.317
Total bilirubin (mg/dL)	28	0.74 ± 0.43	0.57 ± 0.28	−0.17	0.066–0.846	−23.0	0.022

Notes. Mean ± SD; Δ = Post − Pre; %Δ = 100 × (Post − Pre)/Pre. 95% CI for d (Cohen). *p*: paired test (Student’s *t*-test or Wilcoxon) depending on distribution. Sex-stratified analysis was descriptive (no direct comparison between sexes). Rounding note. Δ and %Δ were calculated from non-rounded means and then rounded (Δ to 2 decimals; %Δ to 1). Due to rounding, discrepancies of up to ±1.0 percentage point may occur when recalculating with displayed values. Abbreviations. GGT = γ-glutamyltransferase; SD = standard deviation.

#### 3.2.3. Renal Function and Metabolites

In the total cohort, creatinine decreased by 11.1% (Δ = −0.11 mg/dL; *p* < 0.001; 95% CI for d: 0.206–0.604), accompanied by a parallel 11.0% increase in eGFR (Δ = +8.73 mL/min/1.73 m^2^; *p* < 0.001; 95% CI for d: 0.353–0.764). Urea decreased by 9.5% (Δ = −3.93 mg/dL; *p* < 0.001; 95% CI for d: 0.281–0.685). Uric acid decreased by 13.2% (Δ = −0.95 mg/dL; *p* < 0.001; 95% CI for d: 0.418–0.836). Magnesium showed no significant change, with a small shift compatible with no effect (Δ = −0.028 mg/dL; −1.4%; *p* = 0.338; 95% CI for d: −0.098–0.285).

**Sex-stratified comparisons of Δ**. Significant differences were observed for urea (*p* = 0.032), creatinine (*p* < 0.001), and uric acid (*p* < 0.001). Women showed greater reductions in creatinine (−12.8% vs. −7.1%) and urea (−10.4% vs. −7.2%); for uric acid, reductions were numerically similar (−13.6% vs. −12.3%), although the formal comparison was statistically significant (*p* < 0.001). No significant sex differences were found for eGFR (*p* = 0.997) or magnesium (*p* = 0.057).

These renal and metabolite changes are summarized for the total cohort in Table 11, with sex-stratified results for women and men in Table 12 and Table 13, respectively.

In women (*n* = 77), creatinine decreased by 12.8% (Δ = −0.12 mg/dL; *p* < 0.001; 95% CI for d: 0.161–0.625), with an 11.8% increase in eGFR (Δ = +9.38 mL/min/1.73 m^2^; *p* < 0.001; 95% CI for d: 0.304–0.783). Urea decreased by 10.4% (Δ = −4.21 mg/dL; *p* < 0.001; 95% CI for d: 0.258–0.732), and uric acid by 13.6% (Δ = −0.93 mg/dL; *p* < 0.001; 95% CI for d: 0.354–0.840). Magnesium showed no significant change (*p* = 0.677; 95% CI for d: −0.176–0.271).

**Table 12 nutrients-17-03559-t012:** Renal function and metabolites—Women (*n* = 77).

Variable	*n* Pairs	Pre (Mean ± SD)	Post (Mean ± SD)	Δ (Post − Pre)	95% CI for d (Cohen)	%Δ	*p*
Urea (mg/dL)	77	40.65 ± 9.64	36.44 ± 7.78	−4.21	0.258–0.732	−10.4	<0.001
Creatinine (mg/dL)	77	0.96 ± 0.33	0.84 ± 0.15	−0.12	0.161–0.625	−12.8	<0.001
eGFR (mL/min/1.73 m^2^)	77	79.38 ± 17.99	88.76 ± 16.41	9.38	0.304–0.783	11.8	<0.001
Uric acid (mg/dL)	77	6.83 ± 1.98	5.90 ± 1.49	−0.93	0.354–0.840	−13.6	<0.001
Magnesium (mg/dL)	77	1.969 ± 0.294	1.954 ± 0.280	−0.015	−0.176–0.271	−0.8	0.677

Notes. Mean ± SD; Δ = Post − Pre; %Δ = 100 × (Post − Pre)/Pre. Effect size (Cohen’s d): presented as magnitude (absolute value); 95% CIs are expressed as positive values. *p*: paired test (Student’s *t*-test or Wilcoxon) depending on distribution. Sex-stratified analysis was descriptive (no direct comparison between sexes). eGFR: CKD-EPI 2021 (race-free). Rounding note. Δ and %Δ were calculated from non-rounded means and then rounded (Δ to 2 decimals; %Δ to 1). Due to rounding, discrepancies of up to ±1.0 percentage point may occur when recalculating with displayed values. Abbreviations. eGFR = estimated glomerular filtration rate; SD = standard deviation.

In men (*n* = 28), creatinine decreased by 7.1% (Δ = −0.08 mg/dL; *p* = 0.004; 95% CI for d: 0.180–0.983), while eGFR increased by 8.6% (Δ = +6.94 mL/min/1.73 m^2^; *p* < 0.001; 95% CI for d: 0.279–1.108). Urea decreased by 7.2% (Δ = −3.18 mg/dL; *p* = 0.027; 95% CI for d: 0.050–0.827), and uric acid by 12.3% (Δ = −1.00 mg/dL; *p* < 0.001; 95% CI for d: 0.288–1.119). Magnesium showed a small, non-significant decrease (Δ = −0.063 mg/dL; −3.0%; *p* = 0.189; 95% CI for d: −0.124–0.629).

**Table 13 nutrients-17-03559-t013:** Renal function and metabolites—Men (*n* = 28).

Variable	*n* Pairs	Pre (Mean ± SD)	Post (Mean ± SD)	Δ (Post − Pre)	95% CI for d (Cohen)	%Δ	*p*
Urea (mg/dL)	28	43.96 ± 10.03	40.79 ± 8.93	−3.18	0.050–0.827	−7.2	0.027
Creatinine (mg/dL)	28	1.17 ± 0.25	1.08 ± 0.21	−0.08	0.180–0.983	−7.1	0.004
eGFR (mL/min/1.73 m^2^)	28	80.62 ± 17.86	87.55 ± 18.90	6.94	0.279–1.108	8.6	<0.001
Uric acid (mg/dL)	28	8.17 ± 1.85	7.16 ± 1.43	−1.00	0.288–1.119	−12.3	<0.001
Magnesium (mg/dL)	28	2.086 ± 0.195	2.022 ± 0.162	−0.063	−0.124–0.629	−3.0	0.189

Notes. Mean ± SD; Δ = Post − Pre; %Δ = 100 × (Post − Pre)/Pre. Effect size (Cohen’s d): presented as magnitude (absolute value); 95% CIs are expressed as positive values. *p*: paired test (Student’s *t*-test or Wilcoxon) depending on distribution. Sex-stratified analysis was descriptive (no direct comparison between sexes). eGFR: CKD-EPI 2021 (race-free). Rounding note. Δ and %Δ were calculated from non-rounded means and then rounded (Δ to 2 decimals; %Δ to 1). Due to rounding, discrepancies of up to ±1.0 percentage point may occur when recalculating with displayed values. Abbreviations. eGFR = estimated glomerular filtration rate; SD = standard deviation.

#### 3.2.4. C-Reactive Protein (CRP)

In the total cohort, standard CRP (non–hs-CRP) decreased by 24.6% (Δ = −1.69 mg/L; *p* < 0.001; 95% CI of d: −0.017–0.368). In sex-stratified comparisons of change magnitude (Δ), the difference was significant (*p* = 0.031), with larger reductions in women (−26.8%) than in men (−12.3%).

These changes in standard CRP are summarized in Table 14, with sex-stratified results for women and men in Table 15 and Table 16, respectively.

In women, standard CRP (non–hs-CRP) showed a non-significant decrease (Δ = −2.14 mg/L; −26.8%; *p* = 0.095; 95% CI for d: −0.034–0.417).

**Table 15 nutrients-17-03559-t015:** Systemic inflammation—Women (*n* = 77).

Variable	*n* Pairs	Pre (Mean ± SD)	Post (Mean ± SD)	Δ (Post − Pre)	95% CI for d (Cohen)	%Δ	*p*
Standard CRP (non–hs-CRP, mg/L)	77	8.00 ± 10.44	5.86 ± 8.90	−2.14	−0.034–0.417	−26.8	0.095

Notes. Mean ± SD; Δ = Post − Pre; %Δ = 100 × (Post − Pre)/Pre. 95% CI for d (Cohen). *p*: paired test (Student’s *t*-test or Wilcoxon) depending on distribution. Sex-stratified analysis was descriptive (no direct comparison between sexes). Rounding note. Δ and %Δ were calculated from non-rounded means and then rounded (Δ to 2 decimals; %Δ to 1). Due to rounding, discrepancies of up to ±1.0 percentage point may occur when recalculating with displayed values. Abbreviations. CRP = C-reactive protein; hs-CRP = high-sensitivity CRP; SD = standard deviation.

In men, standard CRP (non–hs-CRP) showed a small, non-significant decrease (Δ = −0.47 mg/L; −12.3%; *p* = 0.355; 95% CI for d: −0.197–0.550).

**Table 16 nutrients-17-03559-t016:** Systemic inflammation—Men (*n* = 28).

Variable	*n* Pairs	Pre (Mean ± SD)	Post (Mean ± SD)	Δ (Post − Pre)	95% CI for d (Cohen)	%Δ	*p*
Standard CRP (non–hs-CRP, mg/L)	28	3.81 ± 2.36	3.34 ± 3.27	−0.47	−0.197–0.550	−12.3	0.355

Notes. Mean ± SD; Δ = Post − Pre; %Δ = 100 × (Post − Pre)/Pre. 95% CI for d (Cohen). *p*: paired test (Student’s *t*-test or Wilcoxon) depending on distribution. Sex-stratified analysis was descriptive (no direct comparison between sexes). Rounding note. Δ and %Δ were calculated from non-rounded means and then rounded (Δ to 2 decimals; %Δ to 1). Due to rounding, discrepancies of up to ±1.0 percentage point may occur when recalculating with displayed values. Abbreviations. CRP = C-reactive protein; hs-CRP = high-sensitivity CRP; SD = standard deviation.

#### 3.2.5. Thyroid/Hormonal Panel

In the total cohort, TSH showed no significant change (Δ = −0.03 µIU/mL; −1.0%; *p* = 0.935; 95% CI for d: −0.183–0.199). Free T4 showed no significant change (Δ = −0.03 ng/dL; −2.0%; *p* = 0.150; 95% CI for d: −0.051–0.333). Cortisol showed no significant change (Δ = −0.52 µg/dL; −3.8%; *p* = 0.269; 95% CI for d: −0.084–0.300). By contrast, free T3 decreased slightly but significantly (Δ = −0.08 pg/mL; −2.7%; *p* = 0.024; 95% CI for d: 0.029–0.416).

**Sex-stratified comparisons of Δ.** No significant differences were observed for TSH (*p* = 0.245), free T4 (*p* = 0.093), free T3 (*p* = 0.201), or cortisol (*p* = 0.258).

These thyroid and cortisol results are presented for the total cohort in Table 17, with sex-stratified values in Table 18 and Table 19, respectively.

In Women (*n* = 77), TSH, free T4, and cortisol showed no significant change (TSH: *p* = 0.444; 95% CI for d: −0.311–0.136; free T4: *p* = 0.231; 95% CI for d: −0.087–0.362; cortisol: *p* = 0.257; 95% CI for d: −0.095–0.354). Free T3 showed a small, non-significant decrease (Δ = −0.06 pg/mL; −2.0%; *p* = 0.149; 95% CI for d: −0.059–0.391).

**Table 18 nutrients-17-03559-t018:** Thyroid/hormonal panel—Women (*n* = 77).

Variable	*n* Pairs	Pre (Mean ± SD)	Post (Mean ± SD)	Δ (Post − Pre)	95% CI for d (Cohen)	%Δ	*p*
TSH (µIU/mL)	77	2.37 ± 1.92	2.64 ± 2.73	0.27	−0.311–0.136	11.4	0.444
Free T4 (ng/dL)	77	1.24 ± 0.23	1.21 ± 0.17	−0.03	−0.087–0.362	−2.0	0.231
Free T3 (pg/mL)	77	3.15 ± 0.35	3.08 ± 0.41	−0.06	−0.059–0.391	−2.0	0.149
Cortisol (µg/dL)	77	14.02 ± 5.20	13.37 ± 4.66	−0.65	−0.095–0.354	−4.7	0.257

Notes. Mean ± SD; Δ = Post − Pre; %Δ = 100 × (Post − Pre)/Pre. 95% CI for d (Cohen). *p*: paired test (Student’s *t*-test or Wilcoxon) depending on distribution. Sex-stratified analysis was descriptive (no direct between-sex comparison). Rounding note. Δ and %Δ were calculated from non-rounded means and then rounded (Δ to 2 decimals; %Δ to 1). Due to rounding, discrepancies of up to ±1.0 percentage point may occur when recalculating with displayed values. Abbreviations. TSH = thyroid-stimulating hormone; Free T3 = triiodothyronine; Free T4 = thyroxine; SD = standard deviation.

In men (*n* = 28), TSH, free T4, and cortisol showed no significant change (TSH: *p* = 0.247; 95% CI for d: −0.154–0.597; free T4: *p* = 0.437; 95% CI for d: −0.225–0.520; cortisol: *p* = 0.845; 95% CI for d: −0.334–0.407). Free T3 showed a small, borderline, non-significant reduction (Δ = −0.14 pg/mL; −4.3%; *p* = 0.051; 95% CI for d: −0.002–0.767).

**Table 19 nutrients-17-03559-t019:** Thyroid/hormonal panel—Men (*n* = 28).

Variable	*n* Pairs	Pre (Mean ± SD)	Post (Mean ± SD)	Δ (Post − Pre)	95% CI for d (Cohen)	%Δ	*p*
TSH (µIU/mL)	28	3.76 ± 7.41	2.92 ± 3.76	−0.84	−0.154–0.597	−22.4	0.247
Free T4 (ng/dL)	28	1.31 ± 0.25	1.28 ± 0.21	−0.03	−0.225–0.520	−2.3	0.437
Free T3 (pg/mL)	28	3.28 ± 0.46	3.14 ± 0.43	−0.14	−0.002–0.767	−4.3	0.051
Cortisol (µg/dL)	28	12.74 ± 4.63	12.59 ± 3.65	−0.16	−0.334–0.407	−1.2	0.845

Notes. Mean ± SD; Δ = Post − Pre; %Δ = 100 × (Post − Pre)/Pre. 95% CI for d (Cohen). *p*: paired test (Student’s *t*-test or Wilcoxon) depending on distribution. Sex-stratified analysis was descriptive (no direct comparison between sexes). Rounding note. Δ and %Δ were calculated from non-rounded means and then rounded (Δ to 2 decimals; %Δ to 1). Due to rounding, discrepancies of up to ±1.0 percentage point may occur when recalculating with displayed values. Abbreviations. TSH = thyroid-stimulating hormone; Free T3 = triiodothyronine; Free T4 = thyroxine; SD = standard deviation.

For comparability with the literature, FT3 and FT4 are also reported in SI units (pmol/L), together with the FT3/FT4 ratio, using the prespecified conversions in Methods (Section 2.7).

In the total cohort, FT3 decreased slightly but significantly (Δ = −0.13 pmol/L; −2.7%; *p* = 0.024; 95% CI of d: 0.029–0.416). FT4 showed no significant change (Δ = −0.33 pmol/L; −2.0%; *p* = 0.164; 95% CI of d: −0.056–0.329). The FT3/FT4 ratio decreased markedly (Δ = −0.078; −25.1%; *p* < 0.001; 95% CI of d: 0.662–1.114).

**Sex-stratified comparisons (Δ).** No significant differences were observed for any of the three parameters (FT3: *p* = 0.243; FT4: *p* = 0.101; FT3/FT4: *p* = 0.214).

These SI-unit results and FT3/FT4 ratios are summarized for the total cohort in Table 20, with sex-stratified data for women and men in Table 21 and Table 22, respectively.

In women (*n* = 77), FT3 showed a small, non-significant reduction (Δ = −0.10 pmol/L; −2.0%; *p* = 0.149; 95% CI of d: −0.059–0.391). FT4 remained stable (Δ = −0.33 pmol/L; −2.0%; *p* = 0.127; 95% CI of d: −0.050–0.400). The FT3/FT4 ratio decreased significantly (Δ = −0.070; −22.6%; *p* < 0.001; 95% CI of d: 0.541–1.055).

**Table 21 nutrients-17-03559-t021:** Thyroid/Hormonal Panel—Women (*n* = 77).

Variable	*n* Pairs	Pre (Mean ± SD)	Post (Mean ± SD)	Δ (Post − Pre)	95% CI of d (Cohen)	%Δ	*p*
FT3 (pmol/L)	77	4.84 ± 0.53	4.74 ± 0.62	−0.10	−0.059–0.391	−2.0	0.149
FT4 (pmol/L)	77	15.93 ± 2.97	15.60 ± 2.15	−0.33	−0.050–0.400	−2.0	0.127
FT3/FT4	77	0.309 ± 0.070	0.239 ± 0.057	−0.070	0.541–1.055	−22.6	<0.001

Notes. Mean ± SD; Δ = Post − Pre; %Δ = 100 × (Post − Pre)/Pre. 95% CI of d (Cohen). *p*: paired test (Student’s *t*-test or Wilcoxon) depending on distribution. Sex-stratified analysis was descriptive (no direct between-sex comparison). Conversions to SI are described in Section 2.7. Rounding note. Δ and %Δ were calculated from non-rounded means and then rounded (Δ to 2 decimals; %Δ to 1). Due to rounding, discrepancies of up to ±1.0 percentage point may occur when recalculating with displayed values. Abbreviations. FT3 = free triiodothyronine; FT4 = free thyroxine; SD = standard deviation; 95% CI of d (Cohen) = 95% confidence interval for Cohen’s d.

In men (*n* = 28), FT3 showed a borderline decrease (Δ = −0.22 pmol/L; −4.3%; *p* = 0.051; 95% CI of d: −0.002–0.767). FT4 did not change significantly (Δ = −0.34 pmol/L; −2.0%; *p* = 0.844; 95% CI of d: −0.333–0.408). The FT3/FT4 ratio decreased significantly (Δ = −0.100; −32.2%; *p* < 0.001; 95% CI of d: 0.674–1.638).

**Table 22 nutrients-17-03559-t022:** Thyroid/Hormonal Panel—Men (*n* = 28).

Variable	*n* Pairs	Pre (Mean ± SD)	Post (Mean ± SD)	Δ (Post − Pre)	95% CI of d (Cohen)	%Δ	*p*
FT3 (pmol/L)	28	5.04 ± 0.71	4.82 ± 0.66	−0.22	−0.002–0.767	−4.3	0.051
FT4 (pmol/L)	28	16.85 ± 3.26	16.50 ± 2.73	−0.34	−0.333–0.408	−2.0	0.844
FT3/FT4	28	0.311 ± 0.069	0.211 ± 0.052	−0.100	0.674–1.638	−32.2	<0.001

Notes. Mean ± SD; Δ = Post − Pre; %Δ = 100 × (Post − Pre)/Pre. 95% CI of d (Cohen). *p*: paired test (Student’s *t*-test or Wilcoxon) depending on distribution. Sex-stratified analysis was descriptive (no direct between-sex comparison). Conversions to SI are described in Section 2.7. Rounding note. Δ and %Δ were calculated from non-rounded means and then rounded (Δ to 2 decimals; %Δ to 1). Due to rounding, discrepancies of up to ±1.0 percentage point may occur when recalculating with displayed values. Abbreviations. FT3 = free triiodothyronine; FT4 = free thyroxine; SD = standard deviation; 95% CI of d (Cohen) = 95% confidence interval for Cohen’s d.

### 3.3. Associations Between Changes (Δ~Δ)

Pearson correlations were calculated between pre–post changes (Δ = Post − Pre) in metabolic, inflammatory, and body composition variables, using two-tailed tests (α = 0.05). The primary associations of interest were: ΔHOMA-IR~ΔTyG (r = 0.351; *p* < 0.001), ΔHOMA-IR~ΔTrunk fat (kg) (r = −0.163; *p* = 0.097), ΔGGT~ΔTG/HDL-c (r = −0.026; *p* = 0.794), ΔCRP~ΔTrunk fat (kg) (r = 0.010; *p* = 0.917), and ΔRemnant cholesterol~ΔHOMA-IR (r = 0.229; *p* = 0.019). In addition, highly significant correlations were observed for ΔTG/HDL-c~ΔTyG (r = 0.764; *p* < 0.001), ΔRemnant cholesterol~ΔTyG (r = 0.825; *p* < 0.001), and ΔRemnant cholesterol~ΔTG/HDL-c (r = 0.962; *p* < 0.001), underscoring a common axis of glycolipid improvement.

These correlation coefficients are summarized in Table 23.

### 3.4. Safety and Adverse Events

The Adaptive Ketogenic–Mediterranean Protocol (AKMP) was well tolerated; no serious adverse events (SAEs; 0 events) were observed during the 14-week intervention. Throughout follow-up, no relevant changes were detected in participants’ usual medication, and no additional interventions were implemented apart from the AKMP. Safety laboratory analyses revealed no clinically significant adverse alterations. On the contrary, reductions were observed in γ-glutamyl transferase (GGT), alkaline phosphatase (ALP), and bilirubin, together with improvements in urea, creatinine, and eGFR, as well as a decrease in C-reactive protein (CRP), consistent with the data shown in Table 8, Table 9, Table 10, Table 11, Table 12, Table 13, Table 14, Table 15 and Table 16.

## 4. Discussion

We acknowledge that physiological ketosis can arise under several contexts (e.g., fasting/energy deficit, prolonged endurance exercise, and low-carbohydrate feeding). In AKMP, the term “ketogenic” refers specifically to diet-induced (nutritional) ketosis via carbohydrate restriction, within a Mediterranean framework and in line with our previously published theoretical construct.

### 4.1. Glucose–Insulin Axis (Primary Outcome)

In our cohort (*n* = 105; 14 weeks), we observed a rapid, robust improvement in the glucose–insulin axis: HOMA-IR −1.80 (−52.8%; *p* < 0.001), fasting plasma glucose −13.67 mg/dL (−13.9%; *p* < 0.001), and fasting insulin −5.91 µU/L (−44.1%; *p* < 0.001). No medication changes were made during the protocol, which supports the interpretation that improvements were driven by the dietary intervention (AKMP).

The magnitude of effect is consistent with reports on very-low-carbohydrate ketogenic diets (VLCKD). In T2D, Westman et al. documented greater reductions in glucose and insulin at 24 weeks compared with a low-glycemic, hypocaloric diet [45]. In randomized trials, Saslow et al. reported sustained improvements in HbA1c, fasting glucose, and HOMA-IR over 3–12 months, along with reduced medication requirements compared with higher-carbohydrate control diets [32,46]. In a controlled clinical study, Goday et al. confirmed significant improvements in glycemic control and medication reduction in T2D following a clinically supervised VLCKD [47]. In the trials by Tay et al., at 52 weeks [48], 1 year [49] and 2 years [50], glycemic control (HbA1c) with a low-carbohydrate diet was similar (non-inferior) to that of a high-carbohydrate diet, with additional benefits in triglycerides, HDL-c, glycemic variability, and lower medication needs.

Hybrid models provide a key comparative framework. In the Keto-Med trial (crossover, 12 weeks), HbA1c improved similarly under both a well-formulated ketogenic diet (WFKD) and a modified Mediterranean diet (Med-Plus) [14]. In contrast, under the AKMP—anchoring well-formulated ketosis from the outset in a Mediterranean matrix and transitioning to a low-carbohydrate Mediterranean pattern—we reproduced glycemic efficacy (marked reductions in HOMA-IR, glucose, and insulin). The lipid outcomes of Keto-Med and their contrast with AKMP are addressed in Section 4.2 (Lipid profile and composite indices) [14,48].

From a clinical and behavioral physiology perspective, the observed pattern can be explained by complementary mechanisms: (i) correction of a relative carbohydrate excess in the context of reduced energy expenditure; (ii) attenuation of refined sugar and ultra-processed food intake, which engage mesolimbic reward circuits and perpetuate hyperglycemia and hyperinsulinemia; and (iii) the anorexigenic milieu of ketosis, which promotes satiety and blunts the compensatory rise in ghrelin typically observed during weight loss, while also facilitating disengagement from hyperglycemic foods [12,51,52,53,54,55,56,57]. In the AKMP, these mechanisms—common to well-formulated ketogenic diets—are embedded from the outset within an unsaturated Mediterranean framework and culminate in a transition toward a carbohydrate-restricted Mediterranean diet tailored to phenotype and activity level, thereby enhancing clinical sustainability without adversely affecting the lipid profile.

Isocaloric evidence provides a clearer interpretation, as such trials help distinguish the effects of dietary composition from those of weight loss. In a 10-day ketogenic diet study in T2D under stable weight conditions, no improvement in insulin sensitivity was observed, likely due to the short duration and absence of a Mediterranean framework [58]. By contrast, in prediabetes, a 6-month low-carbohydrate intervention centered on whole, healthful foods (without requiring sustained ketosis) reduced HbA1c, glucose, insulin, and HOMA-IR, suggesting that lowering carbohydrate load and improving quality alone may be sufficient to enhance the glucose–insulin axis in real-world practice [59]. This convergence—of carbohydrate load, fat and food quality, and the behavioral component—constitutes the core architecture of the AKMP. Weight-related effects and potential bioenergetic mechanisms are discussed in Section 4.3 (Anthropometry and body composition).

Finally, in the medium to long term, the Mediterranean diet demonstrates good adherence and glycemic benefits, although initial reductions in HOMA-IR and insulin are typically slower and more modest than those achieved with strict ketogenic diets [60,61]. The AKMP proposes a convergence: achieving rapid reductions during the ketogenic phase and then maintaining control with a carbohydrate-adapted Mediterranean pattern rich in EVOO, fish, and fiber, thereby optimizing clinical sustainability without compromising glycemic control.

In summary, our findings—within a framework free of pharmacological adjustments by the study team during the intervention—are consistent with the best available evidence and position the AKMP as an efficient strategy to reverse insulin resistance in populations with obesity and metabolic syndrome.

### 4.2. Lipid Profile and Composite Indices (With Remnant Cholesterol [RC] as the Conditional Co-Primary Outcome)

Under the AKMP, the lipid profile showed consistent improvements: triglycerides (TG) −35.1%, total cholesterol −12.2%, and LDL-c −11.2%, while HDL-c remained stable (*p* = 0.508). Remnant cholesterol (RC), the conditional co-primary outcome, also decreased significantly (−10.64 mg/dL; −35.1%; *p* < 0.001). Among composite markers, three axes stand out for their clinical and mechanistic relevance:**(a)** **Remnant cholesterol (RC = TC − LDL-c − HDL-c; conditional co-primary).**

RC decreased significantly. Genetic evidence supports its causal role in ischemic heart disease [5]; furthermore, in the PREDIMED study, RC ≥ 30 mg/dL was associated with an increased risk of major adverse cardiovascular events (MACE), both when LDL-c ≤ 100 mg/dL (Hazard Ratio, HR 2.69; 95% CI: 1.52–4.75) and when LDL-c > 100 mg/dL (HR 1.89; 95% CI: 1.16–3.08), thereby capturing residual risk beyond LDL-c [62]. Consequently, RC should be considered a priority target in the evaluation of residual risk, serving as a complementary marker to LDL-c and particularly relevant in metabolic syndrome, obesity, and diabetes. An important practical advantage is that it incurs no additional cost: it is directly calculated from the routine lipid profile (TC − LDL-c − HDL-c) and is therefore available in most laboratories [5].

**(b)** 
**TyG index (ln[TG × glucose/2]) and TG/HDL-c ratio.**


The TyG index decreased in parallel with simultaneous reductions in TG and glucose. As TyG has been validated in the literature against the hyperinsulinemic–euglycemic clamp as a marker of insulin resistance [4,6], its reduction reinforces the improvement in the glucose–insulin axis with the AKMP and complements the interpretation of HOMA-IR.

The TG/HDL-c ratio decreased substantially, consistent with attenuation of the atherogenic phenotype (↑ small, dense LDL particles) and improvements in the postprandial lipoprotein milieu [3,63]. This pattern aligns with the RCTs by Tay et al., in which low-carbohydrate diets low in saturated fat achieved lower TG and higher HDL-c without relevant increases in LDL-c compared with higher-carbohydrate control diets, both at 12 and 52 weeks and at 2 years [48,49,50].

Accordingly, TyG decreased by 0.23 (−6.0%; *p* < 0.001), and the TG/HDL-c ratio decreased by 1.21 (−37.3%; *p* < 0.001).

#### Comparison with Hybrid and Ketogenic Models

In the Keto-Med trial (12 weeks), the well-formulated ketogenic diet (WFKD) reduced TG by 16% versus 5% under Med-Plus, but this came at the cost of ↑ LDL-c (+10%) compared with ↓ LDL-c (−5%) in the Mediterranean arm [14]. By contrast, within the AKMP—initiating well-formulated ketosis from the outset within a Mediterranean framework featuring characteristic foods (EVOO, oily fish, nuts) and culminating in a final phase of gradual transition to a Mediterranean diet individually adapted to metabolic conditions and lifestyle—we observed reductions in TG and RC without an increase in LDL-c. This finding aligns with the notion that fat quality modulates the LDL-c response in low-carbohydrate contexts [48,49,50].

Beyond macronutrient percentages, the AKMP explicitly operationalizes Mediterranean food quality—extra-virgin olive oil, oily fish, nuts, and non-starchy vegetables—within ketogenic macronutrient limits. This design emphasizes that metabolic outcomes depend not merely on the ratio of carbohydrate, fat, and protein, but on the quality and origin of these nutrients. Defining a diet by percentages alone can be misleading, since a “55–65% carbohydrate and <30% fat” composition could theoretically be achieved even with refined sugars and ultra-processed foods—clearly inconsistent with the Mediterranean paradigm.

In our approach, fat sources are predominantly unsaturated and plant- or marine-derived, and carbohydrates reintroduced in later phases are fiber-rich and minimally processed. This qualitative profile may help explain the observed reductions in triglycerides and remnant cholesterol without LDL-c elevation, contrasting with some non-Mediterranean ketogenic regimens centered on saturated fats or processed meats.

In our real-world clinical practice, menus are fully individualized and dynamically adapted; they prioritize lean meats, fish, EVOO, avocado, and vegetables, ensuring micronutrient adequacy while maintaining nutritional ketosis during the ketogenic phase and enabling a personalized Mediterranean maintenance thereafter.

Over longer durations, meta-analyses of VLCKD have reported modest increases in LDL-c alongside consistent reductions in TG and increases in HDL-c [9]. However, in prolonged, supervised nutritional ketosis, increases in LDL-c have been accompanied by a shift toward larger, less-dense LDL particles, which may partially attenuate the atherogenic risk associated with elevated LDL-c [64]. Moreover, hybrid ketogenic–Mediterranean approaches (Elhayany; KEMEPHY → Mediterranean; SKMD) have shown reductions in HbA1c, TG, and TC with increases in HDL-c and stable or reduced LDL-c, with good medium-term adherence [65,66,67], a pattern consistent with our findings.

**Analytical considerations.** LDL-c was estimated using validated equations (Friedewald and, where applicable, Sampson), and remnant cholesterol (RC) was calculated as TC − LDL-c − HDL-c [41,42]. The marked reduction in TG observed in the cohort reduces the risk of bias in these estimates; nevertheless, results should be interpreted cautiously in subgroups with very high TG.

In summary, the AKMP positioned glycolipid indices (RC, TyG, TG/HDL-c) within the high-efficacy range reported in the literature, while avoiding the LDL-c increase typically observed with some non-Mediterranean ketogenic approaches. The combination of carbohydrate restriction with an unsaturated Mediterranean fat framework and a structured transition to a carbohydrate-restricted Mediterranean diet provides a parsimonious explanation for the integrated improvement in the glycolipid profile and residual risk.

### 4.3. Anthropometry and Body Composition

Over ≈14 weeks with the AKMP, anthropometric outcomes showed marked reductions: body fat mass −12.06 kg (−26.4%), trunk fat −4.88 kg (−22.2%), body weight −14.85 kg (−14.7%), and BMI −5.45 kg/m^2^ (−14.7%) (105 pairs).

By sex, trunk fat decreased by −20.1% in women and −27.6% in men, with men experiencing slightly greater absolute weight loss. In formal comparisons of change magnitude (Δ), the only significant difference between sexes was body weight (*p* = 0.010).

**Metric of central adiposity**. Trunk fat was expressed in kilograms, as this is the metric provided by the body composition analyzer. This regional measure integrates both subcutaneous and visceral adipose tissue and, although it does not discriminate the visceral/android compartment—more closely linked to cardiometabolic risk—it serves as a clinically useful proxy for central adiposity.

The substantial reduction in trunk fat observed over ≈14 weeks, together with improvements in glycolipid indices, supports a favorable shift in the cardiometabolic profile. Correlations between changes (e.g., ΔHOMA-IR~Δtrunk fat) are addressed specifically in Section 3.3.

Our reductions are consistent with findings from the Spanish Ketogenic Mediterranean Diet (SKMD) at 12 weeks (mean losses of −14.14 kg in body weight and −4.70 kg/m^2^ in BMI) and with the KEMEPHY→Mediterranean protocol at 12 months (maintenance of ~−16.11 kg in body weight and −5.15 kg/m^2^ in BMI without regain), both combining carbohydrate restriction within a Mediterranean framework and structured maintenance phases [66,67].

Across the set of classical comparator RCTs (DIRECT, DIETFITS, POUNDS LOST, and Bazzano), weight loss at 12–24 months was greater with low-carbohydrate and/or Mediterranean diets than with low-fat diets, or convergent when adherence and diet quality—rather than precise macronutrient ratios—were emphasized [15,16,68,69]. In T2D, the RCTs by Tay et al. at 52 weeks and 2 years showed comparable weight loss under hypocaloric conditions, with advantages in TG, HDL-c, and glycemic variability when carbohydrate load was reduced [48,49,50]. In the Keto-Med trial (12 weeks), similar weight loss was reported between WFKD and Med-Plus; detailed analysis of the lipid profile and its comparison with the AKMP is presented in Section 4.2 (Lipid profile and composite indices) [14]. In our context, the adaptive transition of the AKMP is designed to sustain weight loss and metabolic control, integrating the behavioral mechanisms outlined in Section 4.1 [12,51,52,53,54,55,56,57].

Although this study did not measure total energy expenditure (TEE) and was not designed to test a ‘metabolic advantage,’ the literature suggests that after ≈2–3 weeks of adaptation, lower-carbohydrate diets may be associated with modest increases in TEE. In a 20-week randomized trial with protein matched between arms (20% of energy) and weight maintained, Ebbeling et al. observed higher TEE with 20% carbohydrates versus 60%: +209 kcal/d (95% CI: 91–326) by intention-to-treat and +278 kcal/d (144–411) in per-protocol analysis. They also reported a linear gradient of ≈+52 kcal/d per −10% carbohydrate [18]. These findings are consistent with a meta-analysis of 29 controlled-feeding studies which, after ≈2–3 weeks of adaptation, showed an average TEE increase of ≈+135 kcal/d in longer studies and a dose–response of ≈+50 kcal/d per −10% carbohydrate [19]. Methodologically, the 20-week trial estimated TEE by DLW under weight-stable conditions and demonstrated robustness in sensitivity analyses (low sensitivity to food quotient assumptions), while the meta-analysis confirmed that, once duration was accounted for, measurement technique (DLW vs. chamber) did not explain additional heterogeneity [18,19]. In an earlier crossover trial following weight loss, a smaller decline in TEE was also observed with a very-low-carbohydrate diet (−97 kcal/d) compared with a low-fat diet (−423 kcal/d), although protein was not matched in that study [23].

**Mechanistic considerations.** A modest increase in TEE after adaptation could be explained by additional energetic costs (gluconeogenesis and the higher thermic effect of protein) and futile cycles—TAG/FFA lipolysis/re-esterification, Ca^2+^ cycling via the sarco/endoplasmic reticulum Ca^2+^-ATPase (SERCA), and creatine/phosphocreatine cycling [24]. At the mitochondrial level, a shift toward fatty acid oxidation may increase the fraction of electron flux entering the ubiquinone pool through complex II and Electron Transfer Flavoprotein:Ubiquinone Oxidoreductase (ETF:QO) [25,26], in contrast to predominant entry via complex I. This pattern, together with phenomena such as redox slip [27] and proton leak [28], would reduce oxidative phosphorylation efficiency (ATP/O ratio) [29,30]. Consequently, to sustain mitochondrial membrane potential (Δψ) and a given rate of ATP synthesis, cells may need to increase respiratory rate (O_2_ consumption) [28,30]. In adipose tissue, the TAG/FFA cycle also consumes ATP and contributes to energy dissipation [31].

While our body composition data cannot demonstrate these mechanisms directly, they are consistent with them, supporting the rationale for future studies using DLW and metabolic chambers to assess TEE and energy partitioning under the AKMP in independent cohorts of overweight/obese patients with typical comorbidities (type 2 diabetes, metabolic syndrome).

Beyond overall weight, central adiposity carries important clinical implications. Although our metric was trunk fat expressed in kilograms, the literature using DXA and specific visceral fat measures (e.g., PREDIMED-Plus) shows that a hypocaloric Mediterranean diet with physical activity reduces visceral fat and preserves lean mass over 1–3 years [70]. In this regard, the marked reductions in trunk fat achieved with the AKMP in only ≈14 weeks suggest that the protocol condensed, within a short period, body composition benefits that other programs typically achieve over longer durations. Finally, the observed sex gradient—with relatively greater reductions in men—should be interpreted descriptively (without inferring causality), but highlights the adaptability of the protocol to modulate targets according to phenotype and clinical context.

### 4.4. Hepatobiliary Markers (GGT, ALP, Bilirubin)

In clinical practice, GGT, ALP, and total bilirubin are routinely included in liver panels because they help distinguish cholestatic, hepatocellular, and bone patterns [71,72,73]. In our cohort, after ≈14 weeks of intervention, we observed a marked reduction in GGT (Δ = −14.73 U/L; −47.0%; *p* < 0.001), accompanied by a modest but significant decrease in ALP (Δ = −10.73 U/L; −6.9%; *p* < 0.001) and a small yet significant reduction in total bilirubin (Δ = −0.11 mg/dL; −18.2%; *p* < 0.001). The fact that bilirubin remained within the normal range argues against clinically meaningful impairment of biliary excretion, while the combination of reduced GGT without concomitant increases in ALP or bilirubin suggests absence of cholestasis and instead points to an overall improvement in the hepatocellular milieu [71,72,73,74,75,76].

By sex, reductions in GGT and total bilirubin were greater in men than in women (*p* between sexes < 0.001 and *p* = 0.002, respectively), whereas ALP showed no sex difference in Δ (*p* = 0.356). In stratified analyses, ALP decreased in women (−8.5%; *p* < 0.001) and remained stable in men (−2.4%; *p* = 0.317).

Beyond their primary clinical use, these analytes retain epidemiological relevance: GGT has been associated with cardiometabolic risk and mortality [74,75,76], Alkaline phosphatase with total and cardiovascular mortality [77,78], and bilirubin with inverse associations with cardiovascular events [79]. These observations enrich the pathophysiological interpretation of the changes we observed, although they do not justify using these markers as formal cardiovascular risk-stratification tools in routine practice [72].

The direction of changes under the AKMP is consistent with the literature. In long-term programs of supervised nutritional ketosis, reductions in liver enzymes—including Alkaline phosphatase—have been reported alongside sustained metabolic improvements in type 2 diabetes [80,81]. In non-alcoholic fatty liver disease, ketogenic interventions have demonstrated rapid reductions in steatosis and reprogramming of hepatic mitochondrial metabolism [81], while low-carbohydrate diets rich in unsaturated fats have also been shown to reduce hepatic fat [82]. In parallel, Mediterranean patterns enriched with polyphenols have demonstrated favorable effects on intrahepatic fat compared with isocaloric comparators [83].

However, neither in Keto-Med [14] nor in other recent ketogenic diet trials [84] were ALP or GGT reported systematically, making our study one of the few to provide a simultaneous, prespecified analysis of GGT, ALP, and bilirubin within a 12–14-week Mediterranean ketogenic protocol.

The biological plausibility of these findings is supported by convergent mechanisms: carbohydrate restriction reduces de novo lipogenesis and favors fatty acid oxidation, while the Mediterranean framework—centered on extra-virgin olive oil, oily fish, nuts, and polyphenols—helps mitigate oxidative stress and hepatocellular lipotoxicity, in addition to limiting fructose and alcohol intake [14,50,82,83,84,85]. This framework explains the parallel reduction in GGT and the slight decrease in ALP, although the absence of isoenzyme fractionation precludes certainty regarding the hepatic versus bone origin of the latter [73,86].

Taken together, the reduction in GGT accompanied by clinically stable or modestly lower ALP and normal-range total bilirubin supports the hepatobiliary safety of the AKMP in real-world practice and suggests an improvement in hepatometabolic status in parallel with weight loss. Although epidemiological associations with cardiovascular events have been reported [74,75,76,77,78,79], the primary interpretation of these analytes in this context should remain hepatobiliary (and bone-related in the case of ALP) and should not be used for cardiovascular risk stratification [71,72,73].

### 4.5. Renal Function and Uric Acid

#### 4.5.1. Renal Function

In our cohort, creatinine decreased by 11.1% (Δ = −0.11 mg/dL), accompanied by a parallel 11.0% increase in eGFR (Δ = +8.73 mL/min/1.73 m^2^; both *p* < 0.001). This pattern is pathophysiologically plausible in weight-loss interventions, where a fall in creatinine (reduced generation from lean mass) may “functionally” elevate estimated eGFR; hence the recommendation to interpret eGFR increases within the clinical context and, when possible, to corroborate them with cystatin C-based eGFR (CKD-EPI 2021, race-free) [44].

From a safety perspective, RCT evidence at 4–12 months in obesity/T2D shows no renal impairment with low- or very-low-carbohydrate diets compared with higher-carbohydrate or low-fat comparators, with no worsening of eGFR or albuminuria [47,87,88]. Over longer durations, a Mediterranean dietary pattern may attenuate eGFR decline in certain subgroups, as observed in CORDIOPREV [89].

In our cohort, sex differences in the magnitude of change (Δ) were significant for urea (*p* = 0.032), creatinine (*p* < 0.001), and uric acid (*p* < 0.001), with greater reductions in women, whereas no differences were observed for eGFR (*p* = 0.997) or magnesium (*p* = 0.057).

Taken together, our findings (↓ creatinine; ↑ eGFR; ↓ urea) are consistent with a neutral renal safety profile of the AKMP over the short term—more compatible with hemodynamic–metabolic improvement than with pathological hyperfiltration. This interpretation aligns with the reduced glycemic and insulinemic load, decreases in trunk adiposity and low-grade systemic inflammation, and improvements in the lipid profile observed in our cohort (↓HOMA-IR −52.8%, ↓glucose, ↓insulin, ↓TG/HDL-c, ↓remnant cholesterol, ↓CRP, and weight loss) [44,47,87,88,89].

#### 4.5.2. Uric Acid

During the initial weeks of nutritional ketosis, a transient increase in serum uric acid levels can be observed; in the medium term, however, trials do not show sustained changes. Evidence from meta-analyses indicates an overall neutral effect of ketogenic diets on serum uric acid, whereas the DASH diet reduces it [90,91]. In our cohort, uric acid decreased by 13.2% (Δ = −0.95 mg/dL; *p* < 0.001), a trajectory consistent with this pattern and with the Mediterranean framework of the AKMP.

### 4.6. Systemic Inflammation (CRP)

In a conventional blood panel, C-reactive protein (CRP) acts as a sentinel marker of inflammation—i.e., it does not identify the cause but indicates the presence and intensity of the inflammatory process. In the context of obesity and insulin resistance, its clinical utility lies in detecting low-grade metabolic inflammation arising from excess adipose tissue—particularly visceral adiposity—through sustained cytokine release (e.g., IL-6) that stimulates hepatic CRP synthesis [1,2,3].

In our cohort, standard CRP (non–hs-CRP) decreased by 24.6% over ≈14 weeks (Δ = −1.69 mg/L; *p* < 0.001), a pattern consistent with attenuation of low-grade systemic inflammation characteristic of the adiposity–insulin-resistant phenotype under the AKMP. Here, we refrained from analyzing correlations (addressed separately) and focused on the direction of effect shown in Table 14, Table 15 and Table 16. The between-sex difference in the magnitude of change (Δ) was significant (*p* = 0.031), with larger reductions in women (−26.8%; Δ = −2.14 mg/L; *p* = 0.095) than in men (−12.3%; Δ = −0.47 mg/L; *p* = 0.355). We interpret this signal cautiously, given the lack of statistical significance in sex-specific paired analyses.

In the general population, median hs-CRP is ≈0.8 mg/L (90th percentile = 3 mg/L; 99th percentile = 10 mg/L). Thus, values > 3 and <10 mg/L are interpreted as chronic low-grade inflammation, whereas values > 10 mg/L warrant repeat testing to rule out acute processes [92,93]. In our study, standard CRP (non–hs-CRP) was measured, which is less sensitive below 3 mg/L; nevertheless, the observed means (6.88 → 5.18 mg/L) fall well within the analytical range, making the clinical interpretation of low-grade inflammation appropriate in this context. These ranges are clinically relevant because central obesity and insulin resistance are associated with sustained CRP elevations mediated by IL-6 and other cytokines secreted from visceral adipose tissue [94,95,96].

The literature indicates that weight loss is the principal determinant of reductions in (hs-)CRP. In the POUNDS LOST trial, hs-CRP decreased by ~25% at 6 months regardless of macronutrient composition [97], and in the trial by Brinkworth et al. (12 months; VLCKD vs. isocaloric low-fat), CRP fell in both arms with comparable weight loss [10]. Nevertheless, in some contexts an additional benefit of carbohydrate restriction has been observed. For example, in Bazzano’s RCT (12 months; adults with obesity), the reduction in (hs-)CRP was greater with a low-carbohydrate diet than with a low-fat diet (between-group Δ = −15.2 nmol/L; 95% CI: −27.6 to −1.9; *p* = 0.024; ≈ −1.75 mg/L) [69]. In severe obesity, Seshadri reported an additional ~−2 mg/L with low-carbohydrate intake in the subgroup with baseline CRP > 3 mg/L, independent of weight loss [98]. In a 12-week trial in adults with obesity, Ruth et al. found that, with similar weight loss, reductions in hs-CRP were greater with a hypocaloric low-carbohydrate diet [99].

In parallel, the Mediterranean pattern provides an intrinsic anti-inflammatory component: reductions in hs-CRP/IL-6 in metabolic syndrome [100], and even **isocaloric** CRP decreases (−26.1% in 5 weeks) without weight loss [101].

Recent evidence is concordant: ketogenic approaches reduce CRP across different trials [34], whereas the Mediterranean diet is the pattern most consistently associated with lower CRP in the general population [102]. Moreover, a meta-analysis specific to low-carbohydrate diets confirmed a modest but significant reduction in CRP compared with low-fat diets (SMD = −0.10; *p* = 0.03), with part of the effect mediated by weight loss [103].

Taken together, the dual approach of the AKMP—carbohydrate restriction (lower insulinemia and glycemic variability) and the Mediterranean framework (EVOO, oily fish, nuts, polyphenols)—provides a plausible explanation for the CRP reduction observed over ≈14 weeks [10,97,98,99,100,101,102].

### 4.7. Cortisol

Over ≈14 weeks with the AKMP, morning serum cortisol remained stable (Δ = −0.52 µg/dL; −3.8%; *p* = 0.269), with no evidence of sustained hyperactivation of the HPA axis and no relevant sex differences (between-sex difference in Δ: *p* = 0.258) (see Section 3.2.5; Table 17, Table 18 and Table 19). This pattern is consistent with the literature in men undergoing very-low-carbohydrate diets, which describes a transient elevation of resting cortisol during the first ~3 weeks followed by normalization; moreover, macronutrient manipulation may alter tissue glucocorticoid metabolism (↑ 11β-HSD1; ↓ inactivation) without changes in plasma cortisol or its diurnal rhythm [104,105]. In 8-week VLCKD interventions in men, salivary cortisol reductions have also been reported, consistent with HPA adaptation after the initial phase [106]. Over longer-term follow-up (~18 months), Mediterranean patterns rich in polyphenols have shown small decreases in basal cortisol, in parallel with cardiometabolic improvements [107]. Notably, direct comparative trials of ketogenic versus Mediterranean diets have rarely evaluated cortisol (e.g., Keto-Med did not report it), limiting strict comparative inferences in the medium term [14]. Methodologically, a single serum measurement captures the “state level” but not the diurnal slope/CAR (cortisol awakening response); future studies could incorporate salivary profiles.

Taken together, our data support that the AKMP does not adversely alter HPA tone in the short-to-medium term and position cortisol as an accompanying biomarker—rather than a primary mediator—of the metabolic benefits observed [14,104,105,106,107].

### 4.8. Thyroid Panel

In our AKMP cohort (14 weeks, *n* = 105), the thyroid panel showed a pattern consistent with euthyroid adaptation: TSH 2.74 ± 4.16 → 2.72 ± 3.02 µIU/mL (Δ = −0.03; −1.0%; *p* = 0.935), FT4 1.26 ± 0.24 → 1.23 ± 0.18 ng/dL (Δ = −0.03; −2.0%; *p* = 0.150), FT3 3.18 ± 0.38 → 3.10 ± 0.41 pg/mL (Δ = −0.08; −2.7%; *p* = 0.024), and cortisol 13.68 ± 5.07 → 13.16 ± 4.41 µg/dL (Δ = −0.52; −3.8%; *p* = 0.269). No significant sex differences were observed in the magnitude of change: TSH (*p* = 0.245), FT4 (*p* = 0.093), FT3 (*p* = 0.201), and cortisol (*p* = 0.258).

In ketogenic interventions with energy restriction, FT3 reductions are usually greater: in adults with epilepsy treated with a modified Atkins diet for 12 weeks, FT3 decreased by −10.6% and FT4 increased by +12.1% with stable TSH [108], magnitudes substantially greater than the −2.7% FT3 decrease observed under AKMP. In non-ketogenic caloric restriction, the longitudinal study by Agnihothri et al. (12 months) showed total T3 − 9.7% (112.7 → 101.8 ng/dL) with TSH and FT4 stable, and a reduction in the T3:FT4 ratio among those who lost >5% of weight [109]. In POUNDS LOST (24 months; four hypocaloric non-ketogenic diets, 35–65% CH), FT3 changes correlated with Δweight and ΔRMR, and baseline FT3 and FT4 predicted greater weight loss at 6 months [16,110]. Similarly, in a trial of time-restricted eating ± low-carb (3 months, MetS), decreases in T3/FT3, stable or increased T4/FT4, stable or reduced TSH, and a reduced T3/T4 ratio (total hormones) were observed, consistent with euthyroid adaptation [111].

The body of evidence indicates that T3 decline appears consistently in both hypocaloric non-ketogenic and ketogenic protocols, suggesting a primary role of energy balance/fat loss rather than ketosis per se [16,108,109,110]. Within this framework, the AKMP—designed to avoid unnecessary caloric restriction and to prioritize the highest intake compatible with realistic, patient-accepted weight loss—achieved attenuation of the thyroid adaptive signal: FT3 decreased only −2.7%, compared with −10.6% under modified Atkins [108] and ~−9.7% with moderate caloric restriction at 12 months [109].

For comparability, we expressed FT3 and FT4 in pmol/L and calculated the FT3/FT4 ratio per participant (Section 2.7). Under AKMP, FT3 decreased by −0.13 pmol/L (−2.7%; *p* = 0.024), FT4 by −0.33 pmol/L (−2.0%; *p* = 0.164), and the FT3/FT4 ratio declined from 0.309 ± 0.070 to 0.231 ± 0.057 (Δ = −0.078; −25.1%; *p* < 0.001). This reduction in the ratio is consistent with patterns described in other interventions: modified Atkins (FT3 ↓, FT4 ↑) [108], time-restricted eating ± low-carb (total T3/T4 ratio ↓) [111], as well as reports of early and sustained reductions in FT3/FT4 with VLCKD [112] and ratio decreases with prolonged non-ketogenic hypocaloric weight loss [113,114]. Against this background, the −25.1% reduction in the AKMP ratio coexisted with only a small FT3 decrease (−2.7%) and stable FT4, suggesting a moderate, non-pathological euthyroid adaptation [108,109,110,111,112,114,115].

In the reduced-weight state, low-dose leptin administration partially reverses the fall in T3 and the reduction in energy expenditure without altering TSH [116,117]. Low-carbohydrate/ketogenic interventions have shown decreases in leptin and increases in the adiponectin-to-leptin ratio, a marker of improved leptin sensitivity, together with a more insulin-sensitive and anti-inflammatory milieu [23,118,119,120,121,122]. These changes may favor a euthyroid profile with a less pronounced decline in FT3 and FT3/FT4. In line with this, the AKMP strategy—by minimizing unnecessary caloric deficit and optimizing the metabolic context—showed an attenuated FT3 decrease and a moderately reduced FT3/FT4 ratio compared with prolonged hypocaloric or ketogenic protocols [16,23,108,109,110,112,118,119,120,121].

### 4.9. Correlation Analyses (Section 3.3)

Over ≈14 weeks under the AKMP, Pearson correlations were evaluated between changes (Δ = Post − Pre) with *n* = 105 complete pairs, using two-tailed tests (α = 0.05). The results showed: (i) a moderate association between ΔHOMA-IR and ΔTyG (r = 0.351; *p* < 0.001); (ii) no significant association between ΔHOMA-IR and Δtrunk fat (kg) (r = −0.163; *p* = 0.097); (iii) a modest association between Δremnant cholesterol (RC) and ΔHOMA-IR (r = 0.229; *p* = 0.019); (iv) no association between ΔGGT and ΔTG/HDL-c (r = −0.026; *p* = 0.794); (v) no association between ΔCRP and Δtrunk fat (kg) (r = 0.010; *p* = 0.917); and (vi) very strong correlations within the lipid axis: ΔTG/HDL-c~ΔTyG (r = 0.764; *p* < 0.001), ΔRC~ΔTyG (r = 0.825; *p* < 0.001), and ΔRC~ΔTG/HDL-c (r = 0.962; *p* < 0.001). These patterns are interpreted below in light of pathophysiology and the metrics employed.

**Glycolipid axis.** The convergence of ΔHOMA-IR~ΔTyG is expected: TyG and HOMA-IR are complementary surrogates of insulin resistance, validated against the clamp and with good diagnostic accuracy at the population level [4,6,123]. The internal synchrony of ΔTyG–ΔTG/HDL-c–ΔRC (with r = 0.764; 0.825; 0.962) is consistent with a coordinated reduction in TG-rich lipoproteins (VLDL/IDL and remnants), causally implicated in atherosclerotic cardiovascular disease [5,62,124]. Together, this reinforces that the AKMP rapidly impacts the hepatic–glycolipid phenotype.

**Insulin resistance and trunk adiposity.** The lack of significance for ΔHOMA-IR~Δtrunk fat over 14 weeks does not contradict the mean reduction of both; rather, it suggests different response timelines and dependence on the metric used. It is plausible that hepatovisceral improvement (glucose/TG) precedes reductions in trunk fat measured by BIA, which lacks the precision of DXA/MRI for discriminating VAT. Operationally, early benefit may be better captured with TyG, TG/HDL-c, and RC and, when available, apoB as a measure of total atherogenic particle number [5,62,124].

**GGT and lipid profile.** The average reduction in GGT (~−47%) coexisted with the absence of correlation between ΔGGT and ΔTG/HDL-c (r = −0.026; *p* = 0.794). This is plausible: GGT reflects oxidative/hepatocellular stress and has greater prognostic value at the population level than as a short-term lipid response marker in individuals [74,76]. Its mean reduction supports hepatometabolic improvement, whereas TyG, TG/HDL-c, and RC (and apoB where available) are more appropriate for monitoring the lipid axis.

**Systemic inflammation.** Although standard CRP (non–hs-CRP) decreased on average, ΔCRP was not associated with Δtrunk fat (r = 0.010; *p* = 0.917). With standard CRP, sensitivity for low-grade inflammation is reduced, and trunk fat assessed by BIA does not isolate VAT; both factors attenuate the correlation between changes. Future studies should employ hs-CRP and a specific VAT metric to increase power for detecting finer associations [125].

### 4.10. Strengths and Limitations

The main strength of this study is that it represents the first large real-world clinical cohort to evaluate a ketogenic protocol inspired by the Mediterranean diet and specifically designed to address metabolic adaptation through anti-plateau strategies. A key differentiating element is the strict individualization of meal plans according to each patient’s clinical characteristics, weight trajectory, and metabolic and behavioral needs. This places the protocol within the framework of precision nutrition, in which dietary interventions must adapt to interindividual variability to optimize both efficacy and safety. In addition, adherence was verified through rigorous and objective procedures (biweekly weight and urinary ketone checks, periodic capillary ketonemia), reinforcing the internal validity of the findings and providing added value compared with less personalized programs.

Among the limitations, the observational design and lack of a control group restrict causal inference and generalizability. Likewise, due to cost, feasibility, or availability, it was not possible to employ reference (gold-standard) analytical methods such as DXA, the hyperinsulinemic–euglycemic clamp, or advanced lipoprotein profiling. With regard to the liver panel, a technical incident in the laboratory prevented the availability of ALT (GPT) and AST (GOT) values, limiting complete characterization in this domain. Reverse T3 was also not measured, which in some contexts is used as a complementary marker of metabolic adaptation, although it is not a gold-standard measure and is not part of routine blood testing. The reference methods for assessing metabolic adaptation or adaptive thermogenesis are the quantification of energy expenditure using doubly labeled water (DLW) for total free-living energy expenditure and indirect calorimetry for resting energy expenditure—techniques that are costly and were not available in this study. Finally, the follow-up duration (≈14 weeks) does not allow for evaluation of medium- to long-term sustainability, an aspect that should be explored in future studies with controlled designs. Additionally, some comparative trials summarized in Ref. [9] were not strictly isocaloric and seldom verified sustained ketosis biochemically; therefore, ketogenic-specific effects cannot be inferred from those studies, a consideration we have kept in mind when interpreting our findings.

## 5. Conclusions

In a cohort of adults with overweight/obesity (mean BMI ≈ 37 kg/m^2^) managed in real-world clinical practice, the Adaptive Ketogenic–Mediterranean Protocol (AKMP)—which combines nutritional ketosis within a Mediterranean matrix and an anti-plateau algorithm—achieved its primary endpoint over ≈ 14 weeks, with substantial improvement in the glucose–insulin axis (ΔHOMA-IR −52.8%; *p* < 0.001). After surpassing the hierarchical threshold, the conditional co-primary outcome (remnant cholesterol, RC) was confirmed with a −35.1% reduction (*p* < 0.001), consistent with decreases in TyG (−6.0%) and TG/HDL-c (−37.3%), and significant loss of adiposity (body weight −14.7%; trunk fat −22.2%). The safety profile was favorable, with reductions in GGT (−47.0%), improvements in eGFR (+11.0%), decreases in CRP (−24.6%), lower LDL-c (−11.2%), and stable HDL-c, with no serious adverse events. Patterns of change were consistent across sexes, with greater absolute weight loss in men. The thyroid panel remained euthyroid, with a modest FT3 decrease, stable FT4, and an FT3/FT4 ratio without clinically relevant changes—compatible with physiological adaptation to weight loss.

These results were obtained in motivated patients seeking weight loss and health improvement, using only routine laboratory testing available in standard clinical practice.

Clinical implications are straightforward. First, the AKMP represents a feasible tool for Primary Care and Endocrinology in patients with overweight/obesity, grounded in Mediterranean foods (extra-virgin olive oil, oily fish, nuts, non-starchy vegetables) and in simple, accessible biomarkers (HOMA-IR, TyG, TG/HDL-c, RC), which allow monitoring of glycolipid improvement and residual risk without increasing costs. Second, the anti-plateau algorithm, which adjusts protein and energy while maintaining carbohydrate restriction, enhances adherence and potentiates trunk fat loss—factors critical for prolonging clinical success beyond the short term. Third, the observed safety profile, together with the neutral or favorable behavior of hepatic, renal, and inflammatory biomarkers, supports the use of AKMP as a first-line strategy in cardiometabolic management programs that aim for rapid efficacy and safety in everyday clinical practice.

In sum, in real-world patients with overweight/obesity, the AKMP demonstrates integrated efficacy, practicality, and analytical safety over 14 weeks, providing a reproducible and transferable operational model for routine care.

## Figures and Tables

**Figure 1 nutrients-17-03559-f001:**
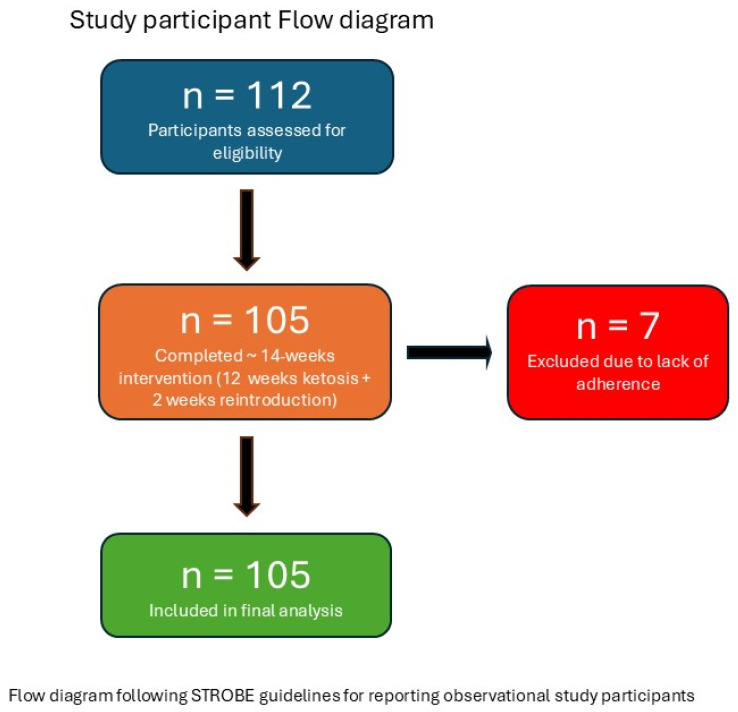
Participant flow diagram, developed in accordance with the STROBE guidelines for observational studies.

**Table 1 nutrients-17-03559-t001:** Baseline cohort characteristics (*n* = 105).

Variable	Baseline Value
Age (years)	46.7 ± 11.4 (range 21–75)
Sex	Women: 77 (73%); Men: 28 (27%)
Weight (kg)	100.84 ± 20.52
BMI (kg/m^2^)	37.00 ± 7.41
Trunk fat (kg)	22.04 ± 4.89 (women 21.84 ± 4.80, *n* = 77; men 22.57 ± 5.19, *n* = 28)

Notes. Unless otherwise specified, data are expressed as mean ± SD; *n* (sample size) is indicated where applicable (e.g., sex-stratified breakdown). BMI = body mass index.

**Table 5 nutrients-17-03559-t005:** Conventional lipids—Total cohort (*n* = 105).

Variable	*n* Pairs	Pre (Mean ± SD)	Post (Mean ± SD)	Δ (Post − Pre)	95% CI for d (Cohen)	%Δ	*p*	*p* Between Sexes (Δ)
TG (mg/dL)	105	151.64 ± 90.84	98.42 ± 38.77	−53.22	0.456–0.880	−35.1	<0.001	0.171
TC (mg/dL)	105	205.92 ± 41.34	180.76 ± 36.31	−25.16	0.653–1.105	−12.2	<0.001	0.928
LDL-c (mg/dL)	105	124.56 ± 34.91	110.63 ± 28.64	−13.93	0.285–0.690	−11.2	<0.001	0.865
HDL-c (mg/dL)	105	51.04 ± 12.49	50.45 ± 10.82	−0.59	−0.127–0.256	−1.2	0.508	0.484

Notes. Mean ± SD; Δ = Post − Pre; %Δ = 100 × (Post − Pre)/Pre. 95% CI for d (Cohen). *p*: paired test (Student’s *t*-test or Wilcoxon) depending on distribution. *p* between sexes (Δ): unpaired comparison of Δ between women and men. LDL-c: calculated using Friedewald (TG < 400 mg/dL) or Sampson (TG ≥ 400 mg/dL) (see Section 2.6). Rounding note. Δ and %Δ were calculated from non-rounded means and then rounded (Δ to 2 decimals; %Δ to 1). Due to rounding, discrepancies of up to ±1.0 percentage point may occur when recalculating with displayed values. Abbreviations. TG = triglycerides; TC = total cholesterol; LDL-c = low-density lipoprotein cholesterol; HDL-c = high-density lipoprotein cholesterol; SD = standard deviation.

**Table 8 nutrients-17-03559-t008:** Liver function—Total cohort (*n* = 105).

Variable	*n* Pairs	Pre (Mean ± SD)	Post (Mean ± SD)	Δ (Post − Pre)	95% CI for d (Cohen)	%Δ	*p*	*p* Between Sexes (Δ)
GGT (U/L)	105	31.36 ± 25.76	16.63 ± 14.32	−14.73	0.537–0.971	−47.0	<0.001	<0.001
Alkaline phosphatase (U/L)	105	156.02 ± 39.65	145.29 ± 43.67	−10.73	0.202–0.599	−6.9	<0.001	0.356
Total bilirubin (mg/dL)	105	0.58 ± 0.34	0.47 ± 0.24	−0.11	0.165–0.560	−18.2	<0.001	0.002

Notes. Mean ± SD; Δ = Post − Pre; %Δ = 100 × (Post − Pre)/Pre. 95% CI for d (Cohen). *p*: paired test (Student’s *t*-test or Wilcoxon) depending on distribution. *p* between sexes (Δ): unpaired comparison of Δ between women and men. Rounding note. Δ and %Δ were calculated from non-rounded means and then rounded (Δ to 2 decimals; %Δ to 1). Due to rounding, discrepancies of up to ±1.0 percentage point may occur when recalculating with displayed values. Abbreviations. GGT = γ-glutamyltransferase; SD = standard deviation.

**Table 11 nutrients-17-03559-t011:** Renal function and metabolites—Total cohort (*n* = 105).

Variable	*n* Pairs	Pre (Mean ± SD)	Post (Mean ± SD)	Δ (Post − Pre)	95% CI for d (Cohen)	%Δ	*p*	*p* Between Sexes (Δ)
Urea (mg/dL)	105	41.53 ± 9.81	37.60 ± 8.29	−3.93	0.281–0.685	−9.5	<0.001	0.032
Creatinine (mg/dL)	105	1.01 ± 0.32	0.90 ± 0.20	−0.11	0.206–0.604	−11.1	<0.001	<0.001
eGFR (mL/min/1.73 m^2^)	105	79.71 ± 17.88	88.44 ± 17.02	8.73	0.353–0.764	11	<0.001	0.997
Uric acid (mg/dL)	105	7.19 ± 2.02	6.24 ± 1.57	−0.95	0.418–0.836	−13.2	<0.001	<0.001
Magnesium (mg/dL)	105	2.000 ± 0.275	1.972 ± 0.255	−0.028	−0.098–0.285	−1.4	0.338	0.057

Notes. Mean ± SD; Δ = Post − Pre; %Δ = 100 × (Post − Pre)/Pre. Effect size (Cohen’s d): presented as magnitude (absolute value); 95% CIs are expressed as positive values. *p*: paired test (Student’s *t*-test or Wilcoxon) depending on distribution. *p* between sexes (Δ): unpaired comparison of the change magnitude (Δ) between women and men. eGFR calculated using CKD-EPI 2021 (race-free). Rounding note. Δ and %Δ were calculated from non-rounded means and then rounded (Δ to 2 decimals; %Δ to 1). Due to rounding, discrepancies of up to ±1.0 percentage point may occur when recalculating with displayed values. Abbreviations. eGFR = estimated glomerular filtration rate; SD = standard deviation.

**Table 14 nutrients-17-03559-t014:** Systemic inflammation—Total cohort (*n* = 105).

Variable	*n* Pairs	Pre (Mean ± SD)	Post (Mean ± SD)	Δ (Post − Pre)	95% CI for d (Cohen)	%Δ	*p*	*p* Between Sexes (Δ)
Standard CRP (non–hs-CRP, mg/L)	105	6.88 ± 9.20	5.18 ± 7.87	−1.69	−0.017–0.368	−24.6	<0.001	0.031

Notes. Mean ± SD; Δ = Post − Pre; %Δ = 100 × (Post − Pre)/Pre. 95% CI for d (Cohen). *p*: paired test (Student’s *t*-test or Wilcoxon) depending on distribution. *p* between sexes (Δ): unpaired comparison of Δ between women and men. Rounding note. Δ and %Δ were calculated from non-rounded means and then rounded (Δ to 2 decimals; %Δ to 1). Due to rounding, discrepancies of up to ±1.0 percentage point may occur when recalculating with displayed values. Abbreviations. CRP = C-reactive protein; hs-CRP = high-sensitivity CRP; SD = standard deviation.

**Table 17 nutrients-17-03559-t017:** Thyroid/hormonal panel—Total cohort (*n* = 105).

Variable	*n* Pairs	Pre (Mean ± SD)	Post (Mean ± SD)	Δ (Post − Pre)	95% CI for d (Cohen)	%Δ	*p*	*p* Between Sexes (Δ)
TSH (µIU/mL)	105	2.74 ± 4.16	2.72 ± 3.02	−0.03	−0.183–0.199	−1.0	0.935	0.245
Free T4 (ng/dL)	105	1.26 ± 0.24	1.23 ± 0.18	−0.03	−0.051–0.333	−2.0	0.15	0.093
Free T3 (pg/mL)	105	3.18 ± 0.38	3.10 ± 0.41	−0.08	0.029–0.416	−2.7	0.024	0.201
Cortisol (µg/dL)	105	13.68 ± 5.07	13.16 ± 4.41	−0.52	−0.084–0.300	−3.8	0.269	0.258

Notes. Mean ± SD; Δ = Post − Pre; %Δ = 100 × (Post − Pre)/Pre. 95% CI for d (Cohen). *p*: paired test (Student’s *t*-test or Wilcoxon) depending on distribution. *p* between sexes (Δ): unpaired comparison of Δ between women and men. Rounding note. Δ and %Δ were calculated from non-rounded means and then rounded (Δ to 2 decimals; %Δ to 1). Due to rounding, discrepancies of up to ±1.0 percentage point may occur when recalculating with displayed values. Abbreviations. TSH = thyroid-stimulating hormone; Free T3 = triiodothyronine; Free T4 = thyroxine; SD = standard deviation. Effect size note. Cohen’s d may be presented as a signed value; its 95% CI can include negative values. The direction of change is interpreted via Δ and %Δ.

**Table 20 nutrients-17-03559-t020:** Thyroid/Hormonal Panel—Total cohort (*n* = 105).

Variable	*n* Pairs	Pre (Mean ± SD)	Post (Mean ± SD)	Δ (Post − Pre)	95% CI of d (Cohen)	%Δ	*p*	*p* Between Sexes (Δ)
FT3 (pmol/L)	105	4.89 ± 0.59	4.76 ± 0.63	−0.13	0.029–0.416	−2.7	0.024	0.243
FT4 (pmol/L)	105	16.17 ± 3.06	15.84 ± 2.34	−0.33	−0.056–0.329	−2.0	0.164	0.101
FT3/FT4	105	0.309 ± 0.070	0.231 ± 0.057	−0.078	0.662–1.114	−25.1	<0.001	0.214

Notes. Mean ± SD; Δ = Post − Pre; %Δ = 100 × (Post − Pre)/Pre. 95% CI of d (Cohen). *p*: paired test (Student’s *t*-test or Wilcoxon) depending on distribution. *p* between sexes (Δ): unpaired comparison of Δ between women and men. Conversions to SI are described in Section 2.7. Rounding note. Δ and %Δ were calculated from non-rounded means and then rounded (Δ to 2 decimals; %Δ to 1). Due to rounding, discrepancies of up to ±1.0 percentage point may occur when recalculating with displayed values. Abbreviations. FT3 = free triiodothyronine; FT4 = free thyroxine; SD = standard deviation; 95% CI of d (Cohen) = 95% confidence interval for Cohen’s d.

**Table 23 nutrients-17-03559-t023:** Correlations between changes (Δ)—Pearson (total cohort).

Variable (Δ~Δ)	r	*p*
HOMA-IR~TyG	0.351	<0.001
HOMA-IR~Trunk fat (kg)	−0.163	0.097
GGT~TG/HDL-c	−0.026	0.794
CRP (mg/L)~Trunk fat (kg)	0.01	0.917
Remnant cholesterol~HOMA-IR	0.229	0.019
TG/HDL-c~TyG	0.764	<0.001
Remnant cholesterol~TyG	0.825	<0.001
Remnant cholesterol~TG/HDL-c	0.962	<0.001

Notes. r = Pearson correlation coefficient between changes (Δ = Post − Pre); *n* = 105 complete pairs; two-tailed *p*-values (α = 0.05). Abbreviations. HOMA-IR = homeostatic model assessment of insulin resistance; TyG = triglyceride–glucose index; TG/HDL-c = triglycerides-to-HDL cholesterol ratio; CRP = C-reactive protein; GGT = γ-glutamyltransferase.

## Data Availability

All data are provided in the article and its Appendix A.

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
