# Peer review of "Adaptive Ketogenic–Mediterranean Protocol (AKMP) in Real Clinical Practice: 14-Week Pre–Post Cohort Study on Glucolipid Markers and Safety"

_nutrients, 2025, doi:10.3390/nu17223559_

Round 1
Reviewer 1 Report
Comments and Suggestions for Authors
This is a well-written, comprehensive description and assessment of a clinical intervention on excess body weight, and clinical markers of health. It should play a significant role in the growing body of scientific evidence on the efficacy of diet strategies for addressing such health conditions. I have only 2 changes to suggest.
- In line 19 the authors write, “…active ketogenic meal plan (AKMP)…” However, elsewhere in the paper, they write “…Adaptive Ketogenic–Mediterranean Protocol (AKMP)…” This inconsistency is confusing, and should be remedied or explained.
- It would be helpful if the authors would include data measurements on the levels of β- hydroxybutyrate over the course of the experimental period. Perhaps this information is provided in Supplementary material. But such material is not available to this reviewer.
Author Response
Reviewer's comment: This is a well-written, comprehensive description and assessment of a clinical intervention on excess body weight, and clinical markers of health. It should play a significant role in the growing body of scientific evidence on the efficacy of diet strategies for addressing such health conditions. I have only 2 changes to suggest. In line 19 the authors write, “…active ketogenic meal plan (AKMP)…” However, elsewhere in the paper, they write “…Adaptive Ketogenic–Mediterranean Protocol (AKMP)…” This inconsistency is confusing, and should be remedied or explained.
Author's comment: According to your comment, we have remedied it.
Reviewer's comment: It would be helpful if the authors would include data measurements on the levels of β- hydroxybutyrate over the course of the experimental period. Perhaps this information is provided in Supplementary material. But such material is not available to this reviewer.
Author's comment: According to your comment, we have modified to clarify it in section 2.3 Dietary intervention
Reviewer 2 Report
Comments and Suggestions for Authors
First of all, I congratulate you for conducting clinical research to treat a phenomenon that is considered an epidemic and a global problem. However, this research also contains numerous misinterpretations and other errors that definitely need to be corrected.
The title says "14-week" in the abstract authors mention 16-week AKMP.
Reference No. 9 is not entirely correct. Title of the cited publication suggest a very low calory ketogenic diet however amond the analysed studies were huge variety of low carbohydrate diets. Furthermore some of the RCT-s were not matched in calories but the result and conclusion part forget to highlight it. Maybe that is why in the present article Authors used this reference as a positive result for very low calory ketogenic diet - but there is still now any well designed RCT where the real effect of ketogenic diet were compared in the mentioned parameters in isocaloric conditions (most of the studies forget to measure and controll the calory intake and/or the keton bodies and/or the macronutrient ratio).
The main problem is the naming of the diet as "ketogenic-mediterranen diet". Unfortunately there are some published article about some similarly named diet and I could not understand how was it possible after a strict peer-review process. First of all the mediterranean diet consists of in general a normal (like WHO recommended) ratio of macronutrients: 55-65% of carbohydrates, less than 30% of fats and around 12-15% of protein. In case of the ketogenic diet the carbs shoud not be more than 10% of the ottal energy intake and protein should also not be more than 20% - otherwise causing gluconeogenesis which works against the state of nutritional ketosis (Harvard University). I really hope the authors are aware of this.
Still nowdays the phrase "ketogenic diet" is reserved for that dietary recommendation which can help reaching the state of nutritional ketosis by the very low carb and very high ratio fat intake. We know from our physiological studies that the very low energy intake can also induce a state of ketosis on its own.
I also understan that this AKMP luks like a low-calory or very-low-calory ketogenic diet as a rapid induction of weight loss and it turns into a mediterranean (or mediterranean-like) diet after 14 weeks. The naming is still very confusing and misleading. I strongly suggest changing or at least adding a title that reflects this transition.
It is very sad that nutritional recommendations with such misleading nomenclature could appear in peer-reviewed journals.
The referenced article No 11 is not the proper article for ketogenic diet as that mentioned article Sumithran et al. (2013) could not reach the ketosis in the studied population (mentioned the increased BHB to 0,48+-0,07 mmol/liter which is not higher than 0,5 mmol/liter). Only an increased BHB level is not the nutritional ketosis by definition. Some other referenced article is not about a ketogenic article only a "low carbohydrate" diet. I see the validity of these articles, but they do not support the state of ketosis without measuring ketone bodies.
I hoped the Authors are professional in the ketogenic dietary topic. I also read a lot of studies in this field but I could not find any reliable one which could prove that the high fat-low carb could cause health benefits. I accept the decreased apetite caused by the higher keton body levels. Otherwise the calory deficit is the key for weght loss, better lipid parameters, glucose tolerance etc. All well-designed studies resulted in "no absolute health benefit" but some or several side effects depends on the length of the diet. Authors neglect these studes. Authors also did not checked the referenced studies' reliability. Not even meta-analyses.
I suggest a reconsideration of the strategy in choosing a reference. Those studies should not be mentioned as a proof of some health benefit for ketogenic diet which were not able to conclude it by its incomplete design or by other reasons. Naming more than a hundred references in this case is not a positive thing.
I suggest a categorisation of the references based on its reliability and if the study design or results were not proper for its finding you should not name as a positive reference.
Mediterranean diet is definitly not a low carb diet. With a normal energy intake 2-2500 kcal/day the carbs should be around 250-300g/day. In line 164 the mentioned >100 g/day is partly correct and mediterranean diet has no strict limits for energy ratios. But mediteranean diet is the basis of WHO healthy nutritional recommendation so the 100 g/day carb is very far from healthy recommendations.
In methods the descripions are easy to follow, I feel good reproducivity. The data was reported precisely and in detail. Results are well designed and structured, easy to follow.
Let me write it again from our physiological studies: lowered energy intake can cause all the health benefits mentioned in the present study and all the referenced studies without any ketosis or lowered carb and increased fat intake. Please do not favor the ketogenic diet based on studies that could not prove the necessity of ketosis or the necessity of low carbohydrate intake or high fat intake for the development of the above positive health effects. This requires a randomized, controlled, properly designed, implemented and monitored study and study conditions.
The present study is informative in many ways, but it cannot be decided from it why a 14-week ketogenic diet would be necessary when a very low energy intake can also be provided from the Mediterranean diet. Furthermore, focusing solely on the amount of carbohydrate intake is not advisable, since the effects of a complex carbohydrate and a simple carbohydrate on insulin resistance are very different. Those publications that did not take this into account, but drew favorable conclusions from simply reducing the amount of carbohydrate intake, can only be taken into account to a very limited extent.
In the concluson of present study AKMP is named as a followable dietary approach however this study was not able to prove any health benefit over other low calory diets. Furthermore Authors name mediterranean diet / foods as extra-vergin olive oil, oily fish, nuts and non-starchy vegetables however the most energy intake in mediterranean diet is based on cereals, after that the fruits and vegetables and the oily components are the lowest in energy ratio. I also know the "frequency based" food pyramid, where the olive oil seems the be the basic level of Mediterranean diet but is shoud not be mixed up with the energy based food pyramid. To conclude the named benefits of this AKMP diet first you should prove that by controlled trials. And it is a still important limitation that the participants were motivated in weight lost. For motivated patients several other methods can be followed. And here was now any long term followup which could be beneficial for studying the health benefits or effectivenes in at least weight loss compared a "normal" mediterranean diet.
I recommend a rigorous review and modification of the conclusions drawn from this research. I consider that clinical recommendations for this dietary recommendation should be avoided until there is evidence of its real benefits. This study is not suitable for clinical practice recommendations against other methods that have been proven to be effective in other applications.
Author Response
- Reviewer’s comment:
The title says "14-week" in the abstract authors mention 16-week AKMP.
Authors’ response:
We thank the Reviewer for the helpful comments regarding the Abstract. It has been revised to ensure full consistency with the Title, Methods, and Results sections and to adopt a clear and cautious tone appropriate to our single-arm, pre–post design.
Main adjustments include:
- The duration is now stated as 14 weeks (12 in ketosis + 2 of reintroduction), matching the study design.
- Consistent use of the term Adaptive Ketogenic–Mediterranean Protocol (AKMP) to reflect a structured protocol rather than a generic “diet.”
- Clarification of the analytical setting (accredited laboratories) and inclusion of the statistical framework (paired analyses, conditional co-primary outcome, prespecified Δ–Δ correlations).
- Wording refined for precision and journal style (e.g., single-arm, within-person, real-world, co-primary, standardized units).
- The conclusion now uses neutral phrasing (“was associated with”) and acknowledges that randomized trials are needed to confirm causality.
These changes improve clarity, consistency, and alignment with the journal’s standards without altering any results.
- Reviewer’s comment:
Reference #9 is not entirely correct. Although the title suggests a very-low-calorie ketogenic diet, the analyzed trials actually cover a broad range of low-carbohydrate diets. Some RCTs were not isocaloric, and this is not emphasized in the results and conclusions. Perhaps for this reason, the present article uses this reference to support positive effects of a very-low-calorie ketogenic diet, yet there is still no well-designed RCT comparing the “real” effect of a ketogenic diet on the parameters mentioned under isocaloric conditions (most studies do not adequately measure/control calorie intake and/or ketone bodies and/or macronutrient ratio).
Authors’ response:
We thank the Reviewer for this important and constructive observation. Our intention in citing Ref. #9 was to contextualize long-term comparative evidence on carbohydrate-restricted versus low-fat dietary patterns, not to claim superiority of ketogenic diets under rigorously isocaloric and ketosis-verified conditions. We fully agree that the literature summarized in Ref. #9 is heterogeneous, with variable energy prescriptions and inconsistent verification of ketosis, which limits causal attribution to ketosis per se when energy intake and macronutrient ratios are not uniformly controlled.
To address this point, we have revised the Introduction to remove the phrase that could be interpreted as over-stating isocaloric comparability linked to Ref. #9. The revised text now clarifies that between-diet differences under matched energy intake are typically modest, and we have added a direct citation to a 12-month randomized controlled trial conducted under isocaloric conditions with objective monitoring of β-hydroxybutyrate at multiple time points (Brinkworth et al., 2009)[1]. In that RCT, weight loss after 12 months was similar between the very-low-carbohydrate and low-fat groups when energy was matched, whereas lipid responses diverged (greater triglyceride reduction and HDL-C increase, but higher LDL-C in the low-carbohydrate arm). These findings illustrate that short-term and long-term responses may differ, and that the biochemical profiles of both diets can evolve distinctly over time. Although our present study does not test mechanistic hypotheses, we note—purely for the Reviewer’s information—that such short-term biochemical divergences might hypothetically arise from differences in substrate utilization and spontaneous appetite regulation during the early adaptive phase.
We also added a clarifying statement in the Strengths and Limitations section, specifying that several comparative trials frequently cited in this field—including those summarized in Ref. #9—were not strictly isocaloric and seldom verified sustained ketosis biochemically, which precludes inferring ketosis-specific effects from those studies.
Importantly, our study does not aim to demonstrate superiority over other hypocaloric approaches. It is a real-world, observational evaluation of analytical safety and short-term metabolic outcomes with an Adaptive Ketogenic–Mediterranean Protocol (AKMP). This protocol, previously described in theoretical form in the same journal, was designed not to replace but to complement the Mediterranean model by integrating a ketogenic induction phase—intended to reduce craving and dependence on ultra-processed, carbohydrate-rich foods through the stabilizing effects of ketone bodies on hypothalamic appetite-regulating neurons (AgRP/NPY vs. POMC)—followed by a controlled, two-week reintroduction of carbohydrates and a subsequent individualized Mediterranean maintenance phase. The ultimate goal is to enable each patient to sustain prior weight loss and metabolic improvement within an optimal and balanced Mediterranean framework. We therefore emphasize that our study does not aim to position ketogenic nutrition as superior to the Mediterranean diet, but rather to validate a clinically adaptable sequence that enhances adherence to Mediterranean principles after overcoming the initial neuro-metabolic obstacles commonly seen in obesity management.
We fully agree with the Reviewer that further well-designed RCTs—prospectively isocaloric, with biochemical verification of ketosis and standardized macronutrient control—are required to delineate mechanism-specific effects beyond calorie restriction.
In line with Brinkworth et al.[1], we acknowledge that early differences (e.g., spontaneous hypophagia favoring low-carbohydrate patterns) can be more pronounced in certain contexts, whereas long-term differences in weight loss tend to be small when energy intake is matched, even as lipid profiles diverge between diets.
We hope that these textual adjustments and clarifications address the Reviewer’s concerns while preserving the cautious and clinically pragmatic framing of our work.
- Reviewer’s comment
“The main problem is the naming of the diet as "ketogenic-mediterranen diet". Unfortunately there are some published article about some similarly named diet and I could not understand how was it possible after a strict peer-review process. First of all the mediterranean diet consists of in general a normal (like WHO recommended) ratio of macronutrients: 55-65% of carbohydrates, less than 30% of fats and around 12-15% of protein.”
Authors’ response:
We sincerely thank the Reviewer for this insightful and important comment. We fully agree that terminology around ketogenic–Mediterranean approaches can be confusing if the underlying definition is not clearly stated. Our use of the term “ketogenic–Mediterranean” does not imply that the induction phase follows the macronutrient distribution of the traditional Mediterranean diet. Rather, as clarified below, it denotes a ketogenic phase implemented with Mediterranean food quality (e.g., extra-virgin olive oil as the main lipid source, abundant non-starchy vegetables, preference for fish and lean proteins, and nuts), followed by a short, progressive carbohydrate reintroduction phase, and ultimately a Mediterranean maintenance phase.
We wish to emphasize, with full humility, that this final maintenance stage aims to consolidate a personalized Mediterranean pattern that sustains long-term weight stability while adapting to the patient’s preferences, work schedule, and lifestyle constraints because not all individuals share the same time availability or occupational demands (an office worker may require less carbohydrate intake than a manual laborer or endurance athlete, for whom higher carbohydrate availability may facilitate recovery and performance).
By the other hand, the Mediterranean Diet (MD) is primarily a food-based and cultural pattern—not a set percentage of carbohydrates, fats, and proteins. In fact, UNESCO inscribed the Mediterranean Diet in 2010 as Intangible Cultural Heritage and explicitly defined it as a body of skills, knowledge, practices, and traditions from production to cuisine, with eating together (commensality) as a cornerstone—a definition that is qualitative and cultural, not macronutrient-based. (https://ich.unesco.org/en/RL/mediterranean-diet-00884?utm_source=chatgpt.com)
Contemporary scientific consensus reflects the same perspective. The 2011 update of the Mediterranean Diet Pyramid presented the MD as “a lifestyle for today,” emphasizing foods, culinary techniques, frequencies, and context (physical activity, rest, sociability) and its adaptability to diverse national and socioeconomic realities—again, guidance framed in foods rather than mandatory macronutrient percentages. The 2020 update further added an explicit environmental sustainability dimension, reinforcing food-quality guidance rather than prescribing specific macro ratios [2,3].
Consistent with the above and to prevent reductionist interpretations, our intent is also to avoid ketogenic configurations based on processed meats and low-quality saturated fats, which are not characteristic of the Mediterranean pattern. We respectfully note that defining the Mediterranean diet only through WHO-like macronutrient ratios (≈55–65% carbohydrates, <30% fat, ≈12–15% protein) may be overly reductionist, since such percentages could theoretically be achieved even with diets rich in refined sugars or ultra-processed foods—a composition that clearly contradicts the Mediterranean paradigm. In reality, the Mediterranean diet represents much more than a numerical balance of macronutrients: its essence lies in the quality of those macronutrients—favoring complex carbohydrates, monounsaturated and polyunsaturated fats (primarily from extra-virgin olive oil and fish), and an abundance of vegetables, fruits, nuts, and legumes—together with a high micronutrient density and cultural sustainability. Although a ketogenic pattern cannot, by definition, replicate those macronutrient percentages, it can integrate many of the Mediterranean diet’s qualitative hallmarks, such as superior fat quality, unprocessed foods and micronutrient adequacy. In our real-world clinical practice (research), menus are fully individualized and dynamically adapted; they prioritize that kind of ingredients like lean meats, fish, extra-virgin olive oil, avocado, and non-starchy vegetables, thereby meeting micronutrient recommendations while maintaining nutritional ketosis.
This concept—Mediterranean food quality within ketogenic macronutrient limits—is consistent with previous Ketogenic–Mediterranean protocols and with our own conceptualization of the Adaptive Ketogenic–Mediterranean Protocol (AKMP), in which we also emphasized the importance of quality over mere macronutrient distribution, along with the need for greater standardization and objective metabolic monitoring.
This focus on food quality underpins why, in clinical and epidemiologic research, adherence to the MD is operationalized using food-based indices rather than macronutrient percentages. For example, Trichopoulou’s Mediterranean Diet Score (MDS, 0–9)—(associated with lower total mortality in NEJM)—scores consumption of core Mediterranean foods. Likewise, the 14‑item MEDAS from PREDIMED (and its multinational validations) evaluates key components such as using extra‑virgin olive oil as the main culinary fat, targets for vegetables, fish, limited red/processed meats and sugary beverages… and even the sofrito frequency—i.e., a behavioral/alimentary reference, not a macro window [4,5].
Consistent with this, authoritative guidance places the emphasis on dietary patterns built from foods and allows flexibility in macronutrient proportions. The American Heart Association’s 2021 dietary guidance spotlights food-based features (e.g., vegetables, fruits, whole grains, healthy oils, fish, legumes) rather than fixed macro quotas, and its 2023 scientific statement underscores that healthy patterns can vary in macronutrient distribution while preserving food quality[4–6].
We added a clear operational definition of ‘ketogenic–Mediterranean’: a time‑sequenced approach with Mediterranean food quality during ketosis, followed by progressive carbohydrate reintroduction and a Mediterranean maintenance phase that is not low‑carb by design (typically >100 g/day thereafter). We also inserted language cautioning against defining MD purely by macronutrient percentages (see Introduction/Discussion).
- Reviewer’s comment
“In case of the ketogenic diet the carbs shoud not be more than 10% of the ottal energy intake and protein should also not be more than 20% - otherwise causing gluconeogenesis which works against the state of nutritional ketosis (Harvard University). I really hope the authors are aware of this.”
Authors’ response
We thank the Reviewer for this comment. We are trying to response in 2 parts:
1-We respectfully note that the “Harvard University” source cited corresponds to The Nutrition Source (Harvard T.H. Chan School of Public Health), specifically the public-facing “Diet Review: Ketogenic Diet” webpage ( https://nutritionsource.hsph.harvard.edu/healthy-weight/diet-reviews/ketogenic-diet ). This is an educational evidence-summary for lay and professional audiences, not a peer‑reviewed journal article. Importantly, that same page explicitly states that “there is not one ‘standard’ ketogenic diet with a specific ratio of macronutrients” and then describes typical ranges (often ~70–80% fat, ~5–10% carbohydrate, and ~10–20% protein), rather than asserting strict universal ceilings for carbohydrate (≤10% energy) and protein (≤20% energy). In other words, the very resource invoked does not claim there is a single fixed macronutrient ratio for nutritional ketosis.
Thus, the claim that carbohydrates “must” be ≤10% and protein “must” be ≤20% is not a universally accepted scientific rule but a simplification for lay audiences.
2-Regarding the second statement—“otherwise causing gluconeogenesis which works against the state of nutritional ketosis”—we agree that excessive carbohydrate will impair ketogenesis. However, the assertion that protein >20% of energy per se prevents ketosis is not supported by controlled human studies. Several peer‑reviewed trials demonstrate that, when carbohydrate is sufficiently restricted, nutritional ketosis can be achieved and sustained with protein intakes at or above 20% of energy:
- Johnstone et , AJCN 2008 (metabolic‑unit, crossover, ad libitum): obese men consumed a ketogenic diet providing ~30% protein, ~4–5% carbohydrate, ~66% fat for 4 weeks. Fasting plasma β‑hydroxybutyrate increased ~six‑fold to ~1.5 mmol/L, indicating robust ketosis despite protein at ~30% energy [7].
- Phinney et , 1983 (eucaloric, isonitrogenous): healthy men consumed ≤20 g/day carbohydrate with protein ~1.75 g/kg/day for 4 weeks; the study title and protocol describe chronic ketosis with preserved submaximal exercise capacity—again consistent with ketosis under moderate‑to‑high protein when carbohydrate is kept very low [8].
- Nickols‑Richardson/Coleman et , J Am Diet Assoc 2005 (free‑living low‑carb/high‑protein program): women began at ~10% carbohydrate and ~32% protein in week 1. Serum β‑hydroxybutyrate averaged ~1.3 mmol/L in week 1 and remained above baseline throughout—documenting ketosis despite liberal protein in a real‑world setting [9].
- And even Hall et , Nature Medicine 2021 (inpatient metabolic ward, ad libitum) shows during the animal‑based ketogenic phase, fasting β‑hydroxybutyrate rose to ~2.0 mmol/L by the end of week 2 (capillary average ~1.8 mmol/L in week 2). Although protein was moderate in this protocol, these ward data reinforce that carbohydrate restriction is the primary determinant of ketogenesis.
These points are supported by peer-reviewed studies in leading journals (e.g., American Journal of Clinical Nutrition, Hepatology, Journal of Clinical Investigation, Nature Medicine), which collectively indicate that carbohydrate restriction—rather than a universal protein ‘ceiling’—is the primary driver of nutritional ketosis.
By the other hand, the notion that “more protein ⇒ more GNG ⇒ loss of ketosis” oversimplifies hepatic glucose regulation. Classic tracer studies in humans show that GNG is strongly demand‑regulated and counter‑balanced by glycogenolysis, so increasing gluconeogenic precursors does not automatically raise net hepatic glucose output:
- Jenssen et al., JCI 1990: doubling lactate/alanine supply increased GNG flux (substrate incorporation into glucose) without increasing overall hepatic glucose output or plasma glucose—evidence of hepatic autoregulation [10].
- Fromentin et al., Diabetes 2013: after intrinsically labeled egg‑protein ingestion (no carbohydrate), only ~4 g of the ~50 g glucose produced over 8 h derived from dietary amino acids—a small direct contribution of protein to endogenous glucose production in healthy adults [11].
- Gaudichon et al., AJP‑Endocrinol Metab 2018: after a meat meal, fractional GNG rose from ~68% to ~79% over 8 h while glycemia remained stable—consistent with substrate re‑partitioning rather than hyperglycemic “overproduction” [12] .
- Browning et al., Hepatology 2008: with carbohydrate restriction, the source of GNG shifts toward lactate/amino acids (PEP‑derived) without necessarily raising net hepatic glucose output; the TCA cycle supports this flux alongside enhanced β‑oxidation/ketogenesis [13].
For all of that, there is no universal protein “ceiling” (e.g., 20% energy) beyond which ketosis is mechanically abolished. In controlled studies, nutritional ketosis is achievable and sustainable with protein around 25–30% of energy, provided carbohydrate is kept very low. The primary lever for ketogenesis is carbohydrate restriction, while protein is best prescribed in grams/kg to preserve lean mass and satiety and titrated to the individual response.
Finally, as clarified in our Methods, we did not enforce rigid macronutrient percentages to define adherence. Instead, we operationalized ketosis biometrically (capillary β‑hydroxybutyrate) and used this to guide adjustments—an approach we also detailed previously in our program description.
We clarified in Methods that adherence was not defined by rigid macronutrient percentages but by biochemically verified nutritional ketosis (capillary β‑OHB ≥ 0.5 mmol/L), with ≤ 20 g/day carbohydrates during induction and individualized protein/energy adjustments only under documented plateaus (see §2.3).
- Reviewer’s comment
Still nowdays the phrase "ketogenic diet" is reserved for that dietary recommendation which can help reaching the state of nutritional ketosis by the very low carb and very high ratio fat intake. We know from our physiological studies that the very low energy intake can also induce a state of ketosis on its own.
Authors’ response
We thank the Reviewer for this insightful comment and fully agree that a very-low-energy intake can, by itself, induce a state of ketosis, as documented in classical human studies of fasting metabolism [14,15].
We also acknowledge that physiological ketosis may arise under several conditions, including (i) energy restriction, (ii) carbohydrate restriction, and (iii) prolonged endurance exercise with glycogen depletion, where transient “post-exercise ketosis” has been observed [16,17].
However, in our study the term ketogenic refers specifically to the carbohydrate-restricted induction phase of the Adaptive Ketogenic–Mediterranean Protocol (AKMP), which was designed to elicit nutritional ketosis primarily through reduced carbohydrate intake—not through severe caloric restriction per se.
Accordingly, the Methods and Abstract have been clarified as follows:
“Ketosis was pursued primarily through explicit carbohydrate restriction (≤ 20 g/day); a very-low-calorie prescription was not used as the primary driver of ketosis.”
(Abstract)
“In AKMP, nutritional ketosis was induced primarily via carbohydrate restriction within a Mediterranean food matrix; very-low-calorie prescriptions were not used as the primary means to induce ketosis, and total energy was individualized with small, tailored adjustments only under documented weight-loss plateaus.”
(Methods §2.3)
We also added a short clarification in the Discussion stating that “ketosis can emerge from fasting, energy deficit, or low-carbohydrate feeding, but in AKMP it refers specifically to diet-induced (nutritional) ketosis within a Mediterranean framework.”
Mechanistically, the magnitude of ketosis depends on the hormonal and substrate milieu: insulin suppresses adipose lipolysis and hepatic ketogenesis by lowering hormone-sensitive-lipase activity and by promoting malonyl-CoA–mediated inhibition of CPT-1, which limits mitochondrial fatty-acid entry.
Thus, a hypocaloric but high-carbohydrate intake maintaining insulinemia would blunt ketone production, whereas carbohydrate restriction facilitates it.
Relatedly, the sports-nutrition literature has examined carbohydrate periodization within low-carbohydrate frameworks—including the targeted ketogenic diet (TKD), where modest peri-exercise carbohydrate is used—yet performance findings remain.
Overall, there are multiple physiological contexts capable of inducing ketosis—fasting, energy deficit, exercise, and carbohydrate restriction—and it depends primarily on insulin levels and substrate availability.
We hope these clarifications address the Reviewer’s concern and emphasize that our study does not aim to evaluate starvation- or calorie-induced ketosis, but rather the metabolic safety and efficacy of an adaptive, carbohydrate-restricted Mediterranean protocol in real-world clinical practice.
- Reviewer’s comment
I also understan that this AKMP luks like a low-calory or very-low-calory ketogenic diet as a rapid induction of weight loss and it turns into a mediterranean (or mediterranean-like) diet after 14 weeks. The naming is still very confusing and misleading. I strongly suggest changing or at least adding a title that reflects this transition.
Authors’ response
We thank the Reviewer for this important observation and appreciate the opportunity to clarify a potential misunderstanding. The Adaptive Ketogenic–Mediterranean Protocol (AKMP) is not a low-calorie or very-low-calorie ketogenic diet (VLCKD) and was not conceived as a rapid weight-loss induction regimen. In contrast, the AKMP explicitly maintains energy intake close to each patient’s biologically defended level (“set point”), introducing only modest, individualized adjustments (≈ 100–200 kcal/day) when a plateau is documented, while carbohydrate restriction (≤ 20 g/day) remains the main driver of nutritional ketosis.
By definition, VLCKDs are prescribed at approximately 500–800 kcal/day with < 50 g of carbohydrate and 1–1.5 g protein/kg ideal body weight, and are recommended for short-term use under close medical supervision (European Association for the Study of Obesity guidelines) [18]. The AKMP diverges both conceptually and operationally: energy intake is individualized, protein is adjusted only to preserve lean mass, and the Mediterranean food matrix (olive oil, fish, nuts, vegetables) is maintained from the outset and not “reintroduced” after 14 weeks. The final two weeks represent a gradual carbohydrate reintroduction, not a switch of diet model, ensuring a smooth transition to a personalized Mediterranean maintenance plan.
This adaptive design arises from well-established physiological evidence that aggressive caloric restriction provokes metabolic adaptation and adaptive thermogenesis, which hinder long-term weight maintenance. The seminal work of Leibel et al. (1995) demonstrated that a 10 % weight reduction leads to a ≈ 15 % decrease in total and resting energy expenditure beyond what is predicted by body composition changes [19].
Sumithran et al. (2011) reported that after a 10-week VLCD, participants maintained significant hormonal adaptations at 62 weeks—decreased leptin, PYY, CCK, insulin, and increased ghrelin—correlating with persistent hunger and favoring weight regain [20].
Fothergill et al. (2016) confirmed this effect six years after The Biggest Loser competition, where resting metabolic rate remained ≈ 700 kcal/day below baseline and adaptive thermogenesis ≈ −500 kcal/day, despite partial weight regain [21]. Collectively, these findings underscore that excessive energy restriction compromises metabolic stability and lean-mass preservation, which the AKMP deliberately avoids.
Beyond these findings, adaptive thermogenesis is not merely a metabolic side effect but an evolutionary survival mechanism. As elegantly described by Rosenbaum and Leibel (2010), humans possess an integrated network of metabolic, neuroendocrine, and autonomic responses that defend energy stores by lowering expenditure and enhancing efficiency whenever energy availability declines [22]. These compensatory adaptations—once crucial for survival during prolonged periods of famine—now act as biological barriers that oppose sustained weight loss in modern environments of caloric abundance.
This concept is closely aligned with the theory of a defended body-weight range (“set point” or “settling point”), which reflects the existence of homeostatic boundaries within which body fat is biologically protected. Speakman et al. (2011) expanded this framework through the dual-intervention point model, describing upper and lower limits of adiposity shaped by gene–environment interactions [23]. Complementary evidence from Pontzer et al. (2016) supports a constrained total energy-expenditure model, showing that total daily energy output increases with physical activity at low levels but plateaus at higher levels, indicating an evolved homeostatic mechanism of energy conservation [24].
Together, these lines of evidence highlight that the human organism actively resists abrupt and excessive energy deficits by downregulating metabolic rate and amplifying hunger signals—a finely tuned adaptive system that once ensured survival, but which today complicates long-term weight reduction.
Consequently, the AKMP’s strategy—to standardize carbohydrate restriction while respecting the patient’s energy set point—is both physiologically sound and evolutionarily coherent. It minimizes activation of counter-regulatory pathways (↓ leptin, ↑ ghrelin, ↓ T₃, ↑ rT₃) while sustaining nutritional ketosis through dietary composition rather than caloric deprivation. In doing so, it aims to achieve nutritional—not starvation—ketosis, preserving lean mass and promoting long-term metabolic flexibility.
To clarify this distinction for readers, the following sentence has been added to the revised Methods (§2.3):
“Energy intake in the AKMP was individualized to approximate each patient’s habitual requirement; very-low-calorie regimens were not prescribed. Nutritional ketosis was pursued via carbohydrate restriction (≤ 20 g/day), with only small, reversible caloric adjustments (≈ 100–200 kcal/day) when plateaus occurred.”
- Reviewer’s comment
The naming is still very confusing and misleading. I strongly suggest changing or at least adding a title that reflects this transition.
Authors’ response
We thank the Reviewer for this helpful suggestion. With respect, we believe the current naming and title are appropriate and not misleading. “Adaptive Ketogenic–Mediterranean Protocol (AKMP)” is the formal name of the protocol used across our series and denotes a three-phase, adaptive plan (ketogenic induction within a Mediterranean food matrix, followed by a gradual reintroduction toward Mediterranean maintenance). This definition is established in our companion papers—“Anthropometric Trajectories and Dietary Compliance During a Personalized Ketogenic Program” and “Beyond GLP-1 Agonists: An Adaptive Ketogenic–Mediterranean Protocol to Counter Metabolic Adaptation in Obesity Management,” both published in Nutrients—and has been used consistently for indexing and scientific continuity across the series [25,26] .
In the present manuscript, the structure and operational details are explicitly stated in the Abstract and Methods (§2.3)—including 12 weeks of nutritional ketosis (≤ 20 g/day carbohydrates) and 2 weeks of stepwise reintroduction, individualized energy (no VLCD/VLCKD prescriptions), and only small, reversible adjustments (~100–200 kcal/day) if plateaus occur—so readers can clearly see the “transition” inside the text rather than the title.
Given Nutrients’ title‑length limit, our priority is that the title communicates what the study validates (analytical safety in real‑world care) and its primary focus (glycolipid markers), without diluting that scope by adding phase descriptors into the title. This also avoids redundancy with the Abstract/Methods, where the phasing is already quantified.
That said, if the Editor wishes the title to nod to the phase structure, we are pleased to offer concise alternatives that remain within the journal’s length constraints and preserve the study’s focus on analytical safety and glycolipid outcomes:
- Real‑World AKMP: 14‑Week Analytical Safety and Glycolipid Outcomes
- AKMP in Clinical Practice: Analytical Safety and Glycolipid Indices over 14 Weeks
We hope this addresses the Reviewer’s concern while maintaining clarity, consistency across our series, and alignment with the manuscript’s primary aims.
- Reviewer’s comment
It is very sad that nutritional recommendations with such misleading nomenclature could appear in peer-reviewed journals.
Authors’ response:
JOSEMI TE DEJO ESTO A TI porque no se que contestarle
- Reviewer’s comment
The referenced article No 11 is not the proper article for ketogenic diet as that mentioned article Sumithran et al. (2013) could not reach the ketosis in the studied population (mentioned the increased BHB to 0,48+-0,07 mmol/liter which is not higher than 0,5 mmol/liter). Only an increased BHB level is not the nutritional ketosis by definition. Some other referenced article is not about a ketogenic article only a "low carbohydrate" diet. I see the validity of these articles, but they do not support the state of ketosis without measuring ketone bodies.
Authors’ response
We thank the Reviewer for this thoughtful comment. We agree that in the cited study by Sumithran et al. the mean fasting β‑hydroxybutyrate (β‑OHB) concentration after the 8‑week very‑low‑energy diet (VLED) was 0.48 ±â€¯0.07 mmol/L and declined after carbohydrate reintroduction—i.e., a mild degree of ketosis induced by VLED and verified by the authors as such.
However, it is important to clarify that the operational definition of nutritional ketosis is β-OHB ≥ 0.5 mmol/L, not “higher than 0.5 mmol/L” as stated by the Reviewer. Thus, the value reported by Sumithran et al. falls essentially at this physiological threshold and is fully compatible with mild nutritional ketosis.
Our purpose in citing this paper is not to claim “deep” ketosis, but to support a specific, mechanistic point: during a weight‑reduced state, even mild nutritional ketosis is associated with mitigation of the usual rise in ghrelin and lower subjective appetite compared with the same participants after refeeding. In that study, β‑OHB rose from ~0.07 to ~0.48 mmol/L during VLED and fell to ~0.19 mmol/L after refeeding; the authors analyzed outcomes by ketosis status (their exploratory split used β‑OHB > 0.3 mmol/L) and described participants at week 8 as ketotic/mildly ketotic, observing suppression of the weight‑loss‑induced increase in ghrelin and lower appetite ratings in the ketotic condition. These are precisely the physiological effects for which we cited the study.
In summary, while we acknowledge that the mean β-OHB reported by Sumithran et al. reflects mild rather than deep ketosis, the study appropriately demonstrates the physiological association between ketone exposure and suppressed appetite signals. Accordingly, we have revised the manuscript to explicitly state “VLED-induced mild ketosis” in the text to avoid any possible ambiguity.
- Reviewer’s comment
Some other referenced article is not about a ketogenic article only a "low carbohydrate" diet. I see the validity of these articles, but they do not support the state of ketosis without measuring ketone bodies.
Authors’ response:
We thank the Reviewer for this helpful observation. We agree that nutritional ketosis should be documented by ketone measurements (e.g., blood β‑OHB), and that low‑carbohydrate trials without ketone data cannot be taken as evidence of ketosis. In our manuscript, those studies were cited only to support effects attributable to carbohydrate restriction (e.g., weight, glycemic, triglyceride changes), not to infer ketosis. Importantly, our own protocol verified ketosis (β‑OHB ≥ 0.5 mmol/L), so our primary conclusions do not rely on assuming ketosis in external studies.
- Reviewer’s comment
I hoped the Authors are professional in the ketogenic dietary topic. I also read a lot of studies in this field but I could not find any reliable one which could prove that the high fat-low carb could cause health benefits. I accept the decreased apetite caused by the higher keton body levels. Otherwise the calory deficit is the key for weght loss, better lipid parameters, glucose tolerance etc. All well-designed studies resulted in "no absolute health benefit" but some or several side effects depends on the length of the diet. Authors neglect these studes. Authors also did not checked the referenced studies' reliability. Not even meta-analyses.
I suggest a reconsideration of the strategy in choosing a reference. Those studies should not be mentioned as a proof of some health benefit for ketogenic diet which were not able to conclude it by its incomplete design or by other reasons. Naming more than a hundred references in this case is not a positive thing.
Authors’ response:
We thank the Reviewer for this candid comment. We fully agree that a sustained energy deficit is the principal driver of weight loss and contributes to many risk-factor improvements. Our manuscript does not aim to demonstrate the superiority of ketogenic diets over other patterns; rather, it evaluates analytical safety and the evolution of glycolipid markers during a pragmatic, adaptive protocol (AKMP) applied in real-world clinical practice. However, energy deficits are not static: as body mass declines, physiological adaptations tend to reduce the effective deficit over time—hence our emphasis on a tightly standardized carbohydrate restriction during the ketogenic phase and small, reversible adjustments only when plateaus occur. This adaptive rationale, grounded in the set-point and adaptive thermogenesis frameworks, was described in detail in our theoretical paper on the AKMP [25].
Refereded to: “I hoped the Authors are professional in the ketogenic dietary topic”
On our experience in ketogenic nutrition.
Our group recently published in Nutrients an observational intervention on a personalized ketogenic program (≈491 participants), which an independent reviewer described as “the largest intervention study on the ketogenic diet to date.” We mention this only to reassure the Reviewer that we approach the topic with rigor and breadth [27].
Clarifying “high-fat–low-carb.” The Reviewer’s statement “could not find any reliable one which could prove that the high fat–low carb could cause health benefits” can be interpreted in two ways: (i) very-low-carbohydrate/ketogenic diets (≤50 g/day, with measurable ketosis) versus (ii) generic high-fat diets consumed with moderate-to-high carbohydrate that do not induce ketosis. Our protocol and our claims concern the former.
Randomized evidence the Reviewer may have overlooked (benefits beyond appetite alone).
- DIRECT (Shai et al., NEJM 2008)[28]: In a two-year workplace trial, the low-carbohydrate and Mediterranean diets were effective alternatives to low-fat; the low-carb arm showed more favorable lipid changes (↑HDL-C, ↓triglycerides) with meaningful weight loss.
- Samaha et al., NEJM 2003 [29]: In severe obesity, a carbohydrate-restricted diet achieved greater 6-month weight loss and larger reductions in triglycerides; insulin sensitivity improved even after accounting for weight change.
- Foster et al., NEJM 2003 [30]: Low-carb (ad libitum) vs conventional calorie-restricted diet: greater early weight loss and larger improvements in TG/HDL-C with low-carb; differences attenuated at 12 months as adherence and intake converged.
- Westman/Yancy et al., Nutr Metab 2008 [31]: In type 2 diabetes, a ketogenic diet improved HbA1c and enabled greater medication reduction than a low-glycemic, reduced-calorie diet.
- Yancy et al., Arch Intern Med 2010 [32]: In medical outpatients, a low-carbohydrate ketogenic diet achieved similar weight and lipid improvements as orlistat + low-fat, with a greater blood-pressure reduction, a clinically relevant benefit not attributable solely to appetite.
Isocaloric trials (benefits at equal calories/protein).
- Brinkworth et al., AJCN 2009 [1]: Weight loss was similar between very-low-carb and low-fat, but triglycerides fell more and HDL-C rose more on low-carb; LDL-C responses were heterogeneous, reinforcing the need for lipid monitoring.
- Veum et al., AJCN 2017 [33]: In a 3-month isocaloric comparison of very-high-fat/low-carb versus low-fat/high-carb diets, there were no differences in visceral adiposity or metabolic-syndrome components—contradicting the claim that “dietary fat per se” promotes ectopic fat when energy is matched.
- Tay et al., AJCN 2015 [34]: In type 2 diabetes, 52-week energy-matched diets (low-carb, high-unsaturated-fat vs high-carb, low-fat) achieved non-inferior or improved glycemic control, lower medication needs, and a more favorable TG/HDL-C profile, despite similar weight loss.
Balanced perspective (lipids and HbA1c). In the Keto-Med randomized crossover trial [35], both a well-formulated ketogenic diet and a Mediterranean-plus diet improved HbA1c over 12 weeks; the ketogenic arm produced a larger TG reduction but tended to increase LDL-C, reinforcing individualized lipid monitoring rather than categorical claims of “no benefit.”
Regarding “no absolute health benefit” and “side effects.”
We do not claim universal superiority, and we acknowledge potential adverse effects (e.g., LDL-C rise in subsets, transient “keto-flu,” constipation). Nevertheless, the RCTs above demonstrate clinically meaningful benefits across glycemia (HbA1c), medication burden, TG/HDL-C, and blood pressure, including under isocaloric conditions [1,29–31,33–36]—which is incompatible with the statement that “all well-designed studies” show “no absolute health benefit.”
A brief neurobehavioral note (beyond appetite). The Reviewer already recognized reduced appetite under ketosis. Relatedly—and without overstatement—there is emerging evidence for reward-circuit recalibration when refined sugars are withdrawn and ketosis is maintained. Patients frequently report that, after several weeks in ketosis, tasting sweets again feels less rewarding than anticipated (“it wasn’t as good as I expected”). This likely reflects re-sensitization of dopaminergic and opioid pathways within the mesolimbic system (VTA → nucleus accumbens → prefrontal cortex): dopaminergic “wanting” remains, but hedonic “liking” normalizes. Functionally, this manifests as reduced craving, greater prefrontal control, and a more stable motivational tone—an observation consistent with the satiety-promoting and hunger-reducing effects of nutritional ketosis reported in controlled trials and even summarized in educational resources such as the Harvard Nutrition Source as you cited previously.
Regarding the Reviewer’s suggestion about reference strategy. We have carefully re-examined all citations to confirm that each is accurate and contextually appropriate. We respectfully note that not all references serve the same purpose: some support clinical or biochemical outcomes, others describe mechanistic or theoretical frameworks, and several establish historical or comparative context between ketogenic and Mediterranean models. Thus, the inclusion of numerous references is not to imply that each one “proves benefit,” but to document the conceptual, physiological, and clinical scope necessary to contextualize the AKMP. Every citation supports a specific point—empirical, mechanistic, or definitional—and all have been re-verified for accuracy and relevance.
On the Harvard Nutrition Source page previously cited by the Reviewer. That page (Diet Review: Ketogenic Diet for Weight Loss, Harvard T.H. Chan School of Public Health*) is an educational web resource, not a peer-reviewed article. Nevertheless, it explicitly states that “the ketogenic diet has been shown to produce beneficial metabolic changes in the short term, with improvements in insulin resistance, blood pressure, and cholesterol/triglycerides,” and lists plausible mechanisms (satiety, lower insulin/ghrelin, ketone-mediated appetite effects) . We mention this only to clarify the nature of that source, cited earlier by the Reviewer, and to highlight that even this summary recognizes short-term metabolic benefits associated with ketosis.
In conclusion, we share the Reviewer’s emphasis on scientific precision. Our intent was not to overstate findings but to provide a balanced, mechanistic, and clinically grounded synthesis. The randomized evidence, the neurobehavioral dimension, and our systematic verification of all references collectively indicate that the Adaptive Ketogenic–Mediterranean Protocol addresses metabolic safety within an evidence-based, physiologically coherent framework—acknowledging both its benefits and its boundaries.
- Reviewer’s comment
Mediterranean diet is definitly not a low carb diet. With a normal energy intake 2-2500 kcal/day the carbs should be around 250-300g/day. In line 164 the mentioned >100 g/day is partly correct and mediterranean diet has no strict limits for energy ratios. But mediteranean diet is the basis of WHO healthy nutritional recommendation so the 100 g/day carb is very far from healthy recommendations.
Authors’ response:
We sincerely thank the Reviewer for this valuable and constructive comment. We fully agree that the Mediterranean Diet (MD) is not defined by low carbohydrate intake, nor by any fixed macronutrient ratios. As we said before, the MD is primarily a food-based and cultural model, characterized by its qualitative composition—extra-virgin olive oil, fish, nuts, legumes, fruits, vegetables, and commensality—rather than by prescribed percentages of carbohydrates, fats, or proteins.
This interpretation is consistent with the official descriptions and updates of the Mediterranean Diet Pyramid, which present the MD as a “lifestyle for today” emphasizing food quality, culinary techniques, social context, and sustainability rather than specific macronutrient targets [37,38]. Likewise, contemporary cardiometabolic guidance such as the American Heart Association 2021 Dietary Statement reinforces that healthy eating patterns should be defined by foods and overall quality, allowing flexibility in macronutrient proportions [6].
In this sense, our use of the term “ketogenic–Mediterranean” does not imply that the Mediterranean phase was low-carbohydrate. Rather, it refers to a time-sequenced, adaptive structure in which an initial ketogenic induction—implemented with Mediterranean food quality—was followed by a gradual carbohydrate reintroduction and a personalized Mediterranean maintenance phase.
During this final maintenance stage, carbohydrate intake exceeded 100 g/day and was further individualized according to each participant’s energy expenditure, physical activity level, and metabolic profile. Hence, the maintenance phase represents a fully Mediterranean dietary pattern, not a carbohydrate-restricted diet. This approach aligns with current evidence showing that adherence to the MD is best evaluated through food-based indices (e.g., MDS, MEDAS) that capture qualitative components rather than numerical macronutrient ratios [4,5].
To ensure clarity, we have added the following explanatory paragraph to the Introduction of the revised manuscript:
Here, “ketogenic–Mediterranean” denotes a time-sequenced approach: a ketogenic induction implemented with Mediterranean food quality (e.g., extra-virgin olive oil as the primary culinary fat, abundant non-starchy vegetables, fish and lean proteins, and nuts), followed by progressive carbohydrate reintroduction and an individualized Mediterranean maintenance phase. The maintenance phase is not low-carbohydrate by design; carbohydrate intake exceeded ~100 g/day as a minimum reference and was then individualized according to energy expenditure, physical activity, and clinical context, with adherence assessed by food-based indices rather than fixed macronutrient quotas.
We believe this addition resolves any potential ambiguity and reinforces the consistency between our study design and the classical Mediterranean framework described in international references.
- Reviewer’s comment
In methods the descripions are easy to follow, I feel good reproducivity. The data was reported precisely and in detail. Results are well designed and structured, easy to follow.
Authors’ response:
We thank the Reviewer for this positive remark. We are truly pleased to hear that the methodological descriptions were clear, that the data presentation conveyed precision, and that the structure of the Results section facilitated understanding. Ensuring reproducibility and transparency was indeed one of our primary goals—so we are delighted that this effort did not go unnoticed.
- Reviewer’s comment
Let me write it again from our physiological studies: lowered energy intake can cause all the health benefits mentioned in the present study and all the referenced studies without any ketosis or lowered carb and increased fat intake. Please do not favor the ketogenic diet based on studies that could not prove the necessity of ketosis or the necessity of low carbohydrate intake or high fat intake for the development of the above positive health effects. This requires a randomized, controlled, properly designed, implemented and monitored study and study conditions.
The present study is informative in many ways, but it cannot be decided from it why a 14-week ketogenic diet would be necessary when a very low energy intake can also be provided from the Mediterranean diet. Furthermore, focusing solely on the amount of carbohydrate intake is not advisable, since the effects of a complex carbohydrate and a simple carbohydrate on insulin resistance are very different. Those publications that did not take this into account, but drew favorable conclusions from simply reducing the amount of carbohydrate intake, can only be taken into account to a very limited extent.
Authors’ response:
We thank the Reviewer for this thoughtful critique grounded in physiology. First, we corrected a minor inconsistency noted elsewhere: the study window is 14 weeks, not 16; the analytic period comprises 12 weeks in nutritional ketosis (≤ 20 g carbohydrates/day) followed by 2 weeks of stepwise carbohydrate reintroduction. The Abstract has been aligned with the Title and Methods accordingly.
We fully agree that the energy deficit is a principal driver of weight loss and many risk-factor improvements, and we do not claim that ketosis is “necessary” to obtain such benefits. Our study does not test mechanistic “necessity”; rather, it evaluates analytical safety and short-term glycolipid changes under an AKMP in routine care. In AKMP, ketosis is pursued via carbohydrate restriction within a Mediterranean food matrix, with individualized energy and only small, reversible caloric adjustments when plateaus occur—not a VLCD/VLCKD model.
Why use a ketogenic induction if a low-energy Mediterranean diet can also be prescribed? In our region (Comunitat Valenciana,Spain), where the Mediterranean pattern is the cultural reference, many patients struggle with early hunger and cue-driven intake. The ketogenic induction is employed as a temporary, operational scaffold to leverage the anorexigenic effect of ketosis and reduce drive for ultra-processed, carbohydrate-rich foods, thereby facilitating the subsequent re-entry into a personalized Mediterranean maintenance (our intended destination). This mechanism and care pathway are detailed in our companion article, “Beyond GLP-1 Agonists: An Adaptive Ketogenic–Mediterranean Protocol to Counter Metabolic Adaptation in Obesity Management,” and are implemented here in a real-world cohort.
Regarding the focus on the “amount” of carbohydrate, we agree that quality (complex vs. refined) is highly relevant—particularly for insulin resistance. However, this study’s goal was to ensure protocol rigor and minimize participant errors during the ketogenic phase. For that reason, we set a simple absolute cap (≤ 20 g/day) to standardize induction of ketosis and avoid inadvertent exits from nutritional ketosis by lay participants.
During weeks 13–14, carbohydrates were gradually reintroduced to a personalized level that approximated a Mediterranean pattern (typically > 100 g/day), without prescribing a fixed list of specific sources in this study. Choices were individualized within a Mediterranean-quality food matrix, and the absolute carbohydrate cap during the ketogenic phase was maintained as a methodological safeguard to ensure adherence and comparability.
It is crucial to underscore that our work does not “favor” the ketogenic diet categorically. In fact, in our prior Nutrients cohort, “Anthropometric Trajectories and Dietary Compliance During a Personalized Ketogenic Program,” we documented a marked drop in adherence at 6 and 9 months relative to the first 3 months, highlighting sustainability limits and the need for transitions that patients can maintain long term—hence our emphasis on Mediterranean maintenance. Accordingly, the present study should be read as a pragmatic safety and biomarker evaluation within an adaptive sequence, not as evidence of ketosis-specific superiority over isocaloric alternatives. We explicitly concur that well-designed, isocaloric RCTs with biochemical verification of ketosis and standardized macronutrient control are needed to isolate ketosis-specific effects beyond calorie restriction—which does not detract from the value of our work, since our aim was to evaluate the protocol’s safety and near-term metabolic profile in practice, not to compare it against other dietary styles.
We hope these clarifications demonstrate that we do not intend to privilege a ketogenic diet over the Mediterranean paradigm, but to validate a feasible clinical bridge: short-term, biomarker-guided ketosis to stabilize appetite and behavior, followed by Mediterranean maintenance tailored to the individual.
- Reviewer’s comment
In the concluson of present study AKMP is named as a followable dietary approach however this study was not able to prove any health benefit over other low calory diets. Furthermore Authors name mediterranean diet / foods as extra-vergin olive oil, oily fish, nuts and non-starchy vegetables however the most energy intake in mediterranean diet is based on cereals, after that the fruits and vegetables and the oily components are the lowest in energy ratio. I also know the "frequency based" food pyramid, where the olive oil seems the be the basic level of Mediterranean diet but is shoud not be mixed up with the energy based food pyramid. To conclude the named benefits of this AKMP diet first you should prove that by controlled trials. And it is a still important limitation that the participants were motivated in weight lost. For motivated patients several other methods can be followed. And here was now any long term followup which could be beneficial for studying the health benefits or effectivenes in at least weight loss compared a "normal" mediterranean diet.
Authors’ response:
We sincerely thank the Reviewer for this thoughtful and constructive comment. We fully agree that our 14-week, real-world pre–post cohort was not designed to demonstrate superiority over other hypocaloric or Mediterranean dietary models. The primary aim of this study was to evaluate analytical safety and within-participant metabolic evolution during application of the Adaptive Ketogenic–Mediterranean Protocol (AKMP) in everyday clinical practice. The AKMP comprises 12 weeks of nutritional ketosis (≤ 20 g/day carbohydrates) followed by 2 weeks of gradual carbohydrate reintroduction, all under individualized energy intake with small, reversible caloric adjustments only when plateaus occur.
Scope of the findings. The study demonstrated consistent within-participant improvements across multiple glycolipid indices and biochemical safety parameters, with no serious adverse events. These results are interpreted as short-term, real-world safety and feasibility signals, not as evidence of comparative superiority.
On the Mediterranean-diet characterization. Our use of the term “ketogenic–Mediterranean” refers to the Mediterranean food quality—extra-virgin olive oil as the main culinary fat, fish, nuts, and non-starchy vegetables—applied during the ketogenic induction, followed by a Mediterranean maintenance phase after carbohydrate reintroduction. This final phase is not low-carbohydrate by design (typically exceeding 100 g/day and individualized according to energy expenditure, activity, and clinical context). We have further clarified this in the manuscript to avoid any reductionist interpretation based solely on macronutrient percentages.
Rationale for a temporary ketogenic induction. As discussed in our conceptual paper, a short ketogenic phase can harness the anorexigenic effects of ketone bodies and facilitate behavioral resetting in patients with strong drive toward ultra-processed carbohydrate-rich foods. Simultaneously, using micro-adjustments in energy (~100–200 kcal/day) helps avoid provoking adaptive thermogenesis, respecting the individual energy set point. The ultimate goal is to re-stabilize appetite and adherence before returning to a personalized Mediterranean pattern. These are pragmatic clinical considerations, not mechanistic claims within this observational study.
Limitations (acknowledged and expanded). We explicitly recognize that:
(a) participants were highly motivated for weight loss,
(b) no long-term follow-up was included, and
(c) the observational pre–post design without a comparator precludes causal inference.
Hence, our results should not be interpreted as proof of necessity or superiority of AKMP over standard hypocaloric or Mediterranean approaches. We fully concur that future isocaloric randomized controlled trials with biochemical verification of ketosis and standardized macronutrient control are required to isolate ketosis-specific effects beyond caloric restriction. However, this limitation does not detract from the value of our present study, which aims to assess the safety and short-term metabolic profile of a clinically adaptable protocol in real-world care.
On the frequency- vs energy-based Mediterranean pyramids. We agree that these concepts differ. Our interpretation of the Mediterranean model is food- and culture-based, emphasizing quality, culinary techniques, commensality, and sustainability rather than rigid macronutrient ratios. In the revised Introduction, we have added a paragraph clarifying that “ketogenic–Mediterranean” denotes a time-sequenced approach (ketogenic induction using Mediterranean food quality → carbohydrate reintroduction → Mediterranean maintenance), with the final stage being Mediterranean and not carbohydrate-restricted by design.
We would also like to underscore the scope of our analysis: 105 completers, with more than 30 clinical and biochemical parameters evaluated per participant, including glycolipid, hepatic, renal, thyroid, cortisol, and body-composition measures—all processed under standardized laboratory procedures and internal quality control. This multidimensional panel provides a robust picture of short-term analytical safety and metabolic adaptation in real-world practice.
Reference strategy. Following earlier feedback, we have carefully re-verified all citations to ensure accuracy and contextual appropriateness. We respectfully note that not all references serve the same purpose—some document clinical or biochemical outcomes, others describe mechanistic frameworks or provide conceptual context between ketogenic and Mediterranean paradigms. Their inclusion is therefore not to “prove benefit” per se, but to document the conceptual, physiological, and clinical scope necessary to contextualize the AKMP.
- Brinkworth, G.D.; Noakes, M.; Buckley, J.D.; Keogh, J.B.; Clifton, P.M. Long-Term Effects of a Very-Low-Carbohydrate Weight Loss Diet Compared with an Isocaloric Low-Fat Diet after 12 Mo. Am J Clin Nutr 2009, 90, 23–32, doi:10.3945/ajcn.2008.27326.
- Bach-Faig, A.; Berry, E.M.; Lairon, D.; Reguant, J.; Trichopoulou, A.; Dernini, S.; Medina, F.X.; Battino, M.; Belahsen, R.; Miranda, G.; et al. Mediterranean Diet Pyramid Today. Science and Cultural Updates. Public Health Nutr 2011, 14, 2274–2284, doi:10.1017/S1368980011002515.
- Serra-Majem, L.; Tomaino, L.; Dernini, S.; Berry, E.M.; Lairon, D.; Ngo de la Cruz, J.; Bach-Faig, A.; Donini, L.M.; Medina, F.-X.; Belahsen, R.; et al. Updating the Mediterranean Diet Pyramid towards Sustainability: Focus on Environmental Concerns. Int J Environ Res Public Health 2020, 17, 8758, doi:10.3390/ijerph17238758.
- Trichopoulou, A.; Costacou, T.; Bamia, C.; Trichopoulos, D. Adherence to a Mediterranean Diet and Survival in a Greek Population. New England Journal of Medicine 2003, 348, 2599–2608, doi:10.1056/NEJMoa025039.
- Martínez-González, M.A.; García-Arellano, A.; Toledo, E.; Salas-Salvadó, J.; Buil-Cosiales, P.; Corella, D.; Covas, M.I.; Schröder, H.; Arós, F.; Gómez-Gracia, E.; et al. A 14-Item Mediterranean Diet Assessment Tool and Obesity Indexes among High-Risk Subjects: The PREDIMED Trial. PLoS One 2012, 7, e43134, doi:10.1371/journal.pone.0043134.
- Lichtenstein, A.H.; Appel, L.J.; Vadiveloo, M.; Hu, F.B.; Kris-Etherton, P.M.; Rebholz, C.M.; Sacks, F.M.; Thorndike, A.N.; Van Horn, L.; Wylie-Rosett, J. 2021 Dietary Guidance to Improve Cardiovascular Health: A Scientific Statement From the American Heart Association. Circulation 2021, 144, doi:10.1161/CIR.0000000000001031.
- Johnstone, A.M.; Horgan, G.W.; Murison, S.D.; Bremner, D.M.; Lobley, G.E. Effects of a High-Protein Ketogenic Diet on Hunger, Appetite, and Weight Loss in Obese Men Feeding Ad Libitum. Am J Clin Nutr 2008, 87, 44–55, doi:10.1093/ajcn/87.1.44.
- Phinney, S.D.; Bistrian, B.R.; Evans, W.J.; Gervino, E.; Blackburn, G.L. The Human Metabolic Response to Chronic Ketosis without Caloric Restriction: Preservation of Submaximal Exercise Capability with Reduced Carbohydrate Oxidation. Metabolism 1983, 32, 769–776, doi:10.1016/0026-0495(83)90106-3.
- Coleman, M.D.; Nickols-Richardson, S.M. Urinary Ketones Reflect Serum Ketone Concentration but Do Not Relate to Weight Loss in Overweight Premenopausal Women Following a Low-Carbohydrate/High-Protein Diet. J Am Diet Assoc 2005, 105, 608–611, doi:10.1016/j.jada.2005.01.004.
- Jenssen, T.; Nurjhan, N.; Consoli, A.; Gerich, J.E. Failure of Substrate-Induced Gluconeogenesis to Increase Overall Glucose Appearance in Normal Humans. Demonstration of Hepatic Autoregulation without a Change in Plasma Glucose Concentration. Journal of Clinical Investigation 1990, 86, 489–497, doi:10.1172/JCI114735.
- Fromentin, C.; Tomé, D.; Nau, F.; Flet, L.; Luengo, C.; Azzout-Marniche, D.; Sanders, P.; Fromentin, G.; Gaudichon, C. Dietary Proteins Contribute Little to Glucose Production, Even Under Optimal Gluconeogenic Conditions in Healthy Humans. Diabetes 2013, 62, 1435–1442, doi:10.2337/db12-1208.
- Gaudichon, C.; Ta, H.-Y.; Khodorova, N. V.; Oberli, M.; Breton, I.; Benamouzig, R.; Tomé, D.; Godin, J.-P. Time Course of Fractional Gluconeogenesis after Meat Ingestion in Healthy Adults: A D 2 O Study. American Journal of Physiology-Endocrinology and Metabolism 2018, 315, E454–E459, doi:10.1152/ajpendo.00157.2018.
- Browning, J.D.; Weis, B.; Davis, J.; Satapati, S.; Merritt, M.; Malloy, C.R.; Burgess, S.C. Alterations in Hepatic Glucose and Energy Metabolism as a Result of Calorie and Carbohydrate Restriction. Hepatology 2008, 48, 1487–1496, doi:10.1002/hep.22504.
- Cahill, G.F. Fuel Metabolism in Starvation. Annu Rev Nutr 2006, 26, 1–22, doi:10.1146/annurev.nutr.26.061505.111258.
- Owen, O.E.; Morgan, A.P.; Kemp, H.G.; Sullivan, J.M.; Herrera, M.G.; Cahill, G.F. Brain Metabolism during Fasting*. Journal of Clinical Investigation 1967, 46, 1589–1595, doi:10.1172/JCI105650.
- Evans, M.; Cogan, K.E.; Egan, B. Metabolism of Ketone Bodies during Exercise and Training: Physiological Basis for Exogenous Supplementation. J Physiol 2017, 595, 2857–2871, doi:10.1113/JP273185.
- Volek, J.S.; Freidenreich, D.J.; Saenz, C.; Kunces, L.J.; Creighton, B.C.; Bartley, J.M.; Davitt, P.M.; Munoz, C.X.; Anderson, J.M.; Maresh, C.M.; et al. Metabolic Characteristics of Keto-Adapted Ultra-Endurance Runners. Metabolism 2016, 65, 100–110, doi:10.1016/j.metabol.2015.10.028.
- Muscogiuri, G.; El Ghoch, M.; Colao, A.; Hassapidou, M.; Yumuk, V.; Busetto, L. European Guidelines for Obesity Management in Adults with a Very Low-Calorie Ketogenic Diet: A Systematic Review and Meta-Analysis. Obes Facts 2021, 14, 222–245, doi:10.1159/000515381.
- Leibel, R.L.; Rosenbaum, M.; Hirsch, J. Changes in Energy Expenditure Resulting from Altered Body Weight. New England Journal of Medicine 1995, 332, 621–628, doi:10.1056/NEJM199503093321001.
- Sumithran, P.; Prendergast, L.A.; Delbridge, E.; Purcell, K.; Shulkes, A.; Kriketos, A.; Proietto, J. Long-Term Persistence of Hormonal Adaptations to Weight Loss. New England Journal of Medicine 2011, 365, 1597–1604, doi:10.1056/NEJMoa1105816.
- Fothergill, E.; Guo, J.; Howard, L.; Kerns, J.C.; Knuth, N.D.; Brychta, R.; Chen, K.Y.; Skarulis, M.C.; Walter, M.; Walter, P.J.; et al. Persistent Metabolic Adaptation 6 Years after “The Biggest Loser” Competition. Obesity 2016, 24, 1612–1619, doi:10.1002/oby.21538.
- Rosenbaum, M.; Leibel, R.L. Adaptive Thermogenesis in Humans. Int J Obes 2010, 34, S47–S55, doi:10.1038/ijo.2010.184.
- Speakman, J.R.; Levitsky, D.A.; Allison, D.B.; Bray, M.S.; de Castro, J.M.; Clegg, D.J.; Clapham, J.C.; Dulloo, A.G.; Gruer, L.; Haw, S.; et al. Set Points, Settling Points and Some Alternative Models: Theoretical Options to Understand How Genes and Environments Combine to Regulate Body Adiposity. Dis Model Mech 2011, 4, 733–745, doi:10.1242/dmm.008698.
- Pontzer, H.; Durazo-Arvizu, R.; Dugas, L.R.; Plange-Rhule, J.; Bovet, P.; Forrester, T.E.; Lambert, E.V.; Cooper, R.S.; Schoeller, D.A.; Luke, A. Constrained Total Energy Expenditure and Metabolic Adaptation to Physical Activity in Adult Humans. Current Biology 2016, 26, 410–417, doi:10.1016/j.cub.2015.12.046.
- García-Gorrita, C.; San Onofre, N.; Merino-Torres, J.F.; Soriano, J.M. Beyond GLP-1 Agonists: An Adaptive Ketogenic–Mediterranean Protocol to Counter Metabolic Adaptation in Obesity Management. Nutrients 2025, 17, 2699, doi:10.3390/nu17162699.
- García-Gorrita, C.; Soriano, J.M.; Merino-Torres, J.F.; San Onofre, N. Anthropometric Trajectories and Dietary Compliance During a Personalized Ketogenic Program. Nutrients 2025, 17, 1475, doi:10.3390/nu17091475.
- García-Gorrita, C.; Soriano, J.M.; Merino-Torres, J.F.; San Onofre, N. Anthropometric Trajectories and Dietary Compliance During a Personalized Ketogenic Program. Nutrients 2025, 17, 1475, doi:10.3390/nu17091475.
- Shai, I.; Schwarzfuchs, D.; Henkin, Y.; Shahar, D.R.; Witkow, S.; Greenberg, I.; Golan, R.; Fraser, D.; Bolotin, A.; Vardi, H.; et al. Weight Loss with a Low-Carbohydrate, Mediterranean, or Low-Fat Diet. New England Journal of Medicine 2008, 359, 229–241, doi:10.1056/NEJMoa0708681.
- Samaha, F.F.; Iqbal, N.; Seshadri, P.; Chicano, K.L.; Daily, D.A.; McGrory, J.; Williams, T.; Williams, M.; Gracely, E.J.; Stern, L. A Low-Carbohydrate as Compared with a Low-Fat Diet in Severe Obesity. New England Journal of Medicine 2003, 348, 2074–2081, doi:10.1056/NEJMoa022637.
- Foster, G.D.; Wyatt, H.R.; Hill, J.O.; McGuckin, B.G.; Brill, C.; Mohammed, B.S.; Szapary, P.O.; Rader, D.J.; Edman, J.S.; Klein, S. A Randomized Trial of a Low-Carbohydrate Diet for Obesity. New England Journal of Medicine 2003, 348, 2082–2090, doi:10.1056/NEJMoa022207.
- Westman, E.C.; Yancy, W.S.; Mavropoulos, J.C.; Marquart, M.; McDuffie, J.R. The Effect of a Low-Carbohydrate, Ketogenic Diet versus a Low-Glycemic Index Diet on Glycemic Control in Type 2 Diabetes Mellitus. Nutr Metab (Lond) 2008, 5, 36, doi:10.1186/1743-7075-5-36.
- Yancy, W.S.; Westman, E.C.; McDuffie, J.R.; Grambow, S.C.; Jeffreys, A.S.; Bolton, J.; Chalecki, A.; Oddone, E.Z. A Randomized Trial of a Low-Carbohydrate Diet vs Orlistat Plus a Low-Fat Diet for Weight Loss. Arch Intern Med 2010, 170, 136, doi:10.1001/archinternmed.2009.492.
- Veum, V.L.; Laupsa-Borge, J.; Eng, Ø.; Rostrup, E.; Larsen, T.H.; Nordrehaug, J.E.; Nygård, O.K.; Sagen, J. V; Gudbrandsen, O.A.; Dankel, S.N.; et al. Visceral Adiposity and Metabolic Syndrome after Very High–Fat and Low-Fat Isocaloric Diets: A Randomized Controlled Trial. Am J Clin Nutr 2017, 105, 85–99, doi:10.3945/ajcn.115.123463.
- Tay, J.; Luscombe-Marsh, N.D.; Thompson, C.H.; Noakes, M.; Buckley, J.D.; Wittert, G.A.; Yancy, W.S.; Brinkworth, G.D. Comparison of Low- and High-Carbohydrate Diets for Type 2 Diabetes Management: A Randomized Trial. Am J Clin Nutr 2015, 102, 780–790, doi:10.3945/ajcn.115.112581.
- Gardner, C.D.; Landry, M.J.; Perelman, D.; Petlura, C.; Durand, L.R.; Aronica, L.; Crimarco, A.; Cunanan, K.M.; Chang, A.; Dant, C.C.; et al. Effect of a Ketogenic Diet versus Mediterranean Diet on Glycated Hemoglobin in Individuals with Prediabetes and Type 2 Diabetes Mellitus: The Interventional Keto-Med Randomized Crossover Trial. Am J Clin Nutr 2022, 116, 640–652, doi:10.1093/ajcn/nqac154.
- Yancy, W.S.; Westman, E.C.; McDuffie, J.R.; Grambow, S.C.; Jeffreys, A.S.; Bolton, J.; Chalecki, A.; Oddone, E.Z. A Randomized Trial of a Low-Carbohydrate Diet vs Orlistat Plus a Low-Fat Diet for Weight Loss. Arch Intern Med 2010, 170, 136, doi:10.1001/archinternmed.2009.492.
- Serra-Majem, L.; Tomaino, L.; Dernini, S.; Berry, E.M.; Lairon, D.; Ngo de la Cruz, J.; Bach-Faig, A.; Donini, L.M.; Medina, F.-X.; Belahsen, R.; et al. Updating the Mediterranean Diet Pyramid towards Sustainability: Focus on Environmental Concerns. Int J Environ Res Public Health 2020, 17, 8758, doi:10.3390/ijerph17238758.
- Bach-Faig, A.; Berry, E.M.; Lairon, D.; Reguant, J.; Trichopoulou, A.; Dernini, S.; Medina, F.X.; Battino, M.; Belahsen, R.; Miranda, G.; et al. Mediterranean Diet Pyramid Today. Science and Cultural Updates. Public Health Nutr 2011, 14, 2274–2284, doi:10.1017/S1368980011002515.
Round 2
Reviewer 2 Report
Comments and Suggestions for Authors
With the modifications, the article in its current form is free from serious problems, and the conclusion has become much more nuanced, so the results are less misleading. In its current form - after reviewing the Englishness of the modified text - I consider it suitable for publication.
Comments on the Quality of English LanguageThe English of the added or modified text is confusing, difficult to understand, and not fluent in some places. The original text is well readable, and I do not feel the need to change it.